# Functional Characterization of Mouse and Human Arachidonic Acid Lipoxygenase 15B (ALOX15B) Orthologs and of Their Mutants Exhibiting Humanized and Murinized Reaction Specificities

**DOI:** 10.3390/ijms241210046

**Published:** 2023-06-12

**Authors:** Kumar R. Kakularam, Miquel Canyelles-Niño, Xin Chen, José M. Lluch, Àngels González-Lafont, Hartmut Kuhn

**Affiliations:** 1Department of Biochemistry, Charité–Universitätsmedizin Berlin, Corporate Member of Freie Universität Berlin and Humboldt Universität zu Berlin, D-10117 Berlin, Germany; kumar.1416@gmail.com (K.R.K.); xin.chen2@charite.de (X.C.); 2Departament de Química, Universitat Autònoma de Barcelona, 08193 Barcelona, Spain; miquel.canyelles@uab.cat (M.C.-N.); josemaria.lluch@uab.cat (J.M.L.); angels.gonzalez@uab.cat (À.G.-L.); 3Arquebio S.L., 08005 Barcelona, Spain; 4Institut de Biotecnologia i de Biomedicina (IBB), Universitat Autònoma de Barcelona, 08193 Barcelona, Spain

**Keywords:** eicosanoids, lipids, metabolism, fatty acids, inflammation, atherosclerosis

## Abstract

The arachidonic acid lipoxygenase 15B (ALOX15B) orthologs of men and mice form different reaction products when arachidonic acid is used as the substrate. Tyr603Asp+His604Val double mutation in mouse arachidonic acid lipoxygenase 15b humanized the product pattern and an inverse mutagenesis strategy murinized the specificity of the human enzyme. As the mechanistic basis for these functional differences, an inverse substrate binding at the active site of the enzymes has been suggested, but experimental proof for this hypothesis is still pending. Here we expressed wildtype mouse and human arachidonic acid lipoxygenase 15B orthologs as well as their humanized and murinized double mutants as recombinant proteins and analyzed the product patterns of these enzymes with different polyenoic fatty acids. In addition, in silico substrate docking studies and molecular dynamics simulation were performed to explore the mechanistic basis for the distinct reaction specificities of the different enzyme variants. Wildtype human arachidonic acid lipoxygenase 15B converted arachidonic acid and eicosapentaenoic acid to their 15-hydroperoxy derivatives but the Asp602Tyr+Val603His exchange murinized the product pattern. The inverse mutagenesis strategy in mouse arachidonic acid lipoxygenase 15b (Tyr603Asp+His604Val exchange) humanized the product pattern with these substrates, but the situation was different with docosahexaenoic acid. Here, Tyr603Asp+His604Val substitution in mouse arachidonic acid lipoxygenase 15b also humanized the specificity but the inverse mutagenesis (Asp602Tyr+Val603His) did not murinize the human enzyme. With linoleic acid Tyr603Asp+His604Val substitution in mouse arachidonic acid lipoxygenase 15b humanized the product pattern but the inverse mutagenesis in human arachidonic acid lipoxygenase 15B induced racemic product formation. Amino acid exchanges at critical positions of human and mouse arachidonic acid lipoxygenase 15B orthologs humanized/murinized the product pattern with C20 fatty acids, but this was not the case with fatty acid substrates of different chain lengths. Asp602Tyr+Val603His exchange murinized the product pattern of human arachidonic acid lipoxygenase 15B with arachidonic acid, eicosapentaenoic acid, and docosahexaenoic acid. An inverse mutagenesis strategy on mouse arachidonic acid lipoxygenase 15b (Tyr603Asp+His604Val exchange) did humanize the reaction products with arachidonic acid and eicosapentaenoic acid, but not with docosahexaenoic acid.

## 1. Introduction

In the human genome, six different arachidonate lipoxygenase (*ALOX*) genes have been detected [1] and these genes encode for six functionally distinct ALOX isoforms). In the reference genome of mice, an orthologous enzyme exists for each human ALOX isoform, but in addition a functional *Aloxe12* gene [1,2] encoding for the Aloxe12 has been identified [3]. In humans, the *ALOXE12* gene is a corrupted pseudogene [1,2]. The human *ALOX15B* gene is located in a common *ALOX* gene cluster present on the short arm of chromosome 17 [2]. The mouse *Alox15b* gene is located in a similar *Alox* gene cluster that is localized in a synthetic region on chromosome 11 [1,2].

ALOX-isoforms are lipid peroxidizing enzymes that convert polyunsaturated fatty acids to the corresponding hydroperoxides [4,5]. In cellular systems these peroxides are rapidly reduced to the corresponding alcohols [6]. Alternatively, the peroxides are further converted to more complex secondary ALOX products such as pro-inflammatory leukotrienes [7] or anti-inflammatory resolvins, maresins, and protectins [8,9], which are collectively named specialized pro-resolving mediators (SPMs). Although the biosynthetic pathways of SPMs are rather diverse and although the SPM-induced intracellular signaling mechanisms are far from clear [9], these compounds exhibit a large number of interesting bioactivities [9]. The majority of ALOX isoforms prefer free polyenoic fatty acids as the substrate and under in vivo conditions the formation of ALOX products is limited by the liberation of the polyenoic fatty acids (PUFAs) from cellular ester lipids [10]. However, some isoforms, in particular human ALOX15 and human ALOX15B accept esterified polyenoic fatty acids as the substrate even if they are presented in complex lipid-protein-assemblies, such as nanodiscs [11,12], biomembranes [13,14,15], and different classes of lipoproteins [16,17,18].

In 1997, human ALOX15B was first described as AA 15-lipoxygenating enzyme, which is high level expressed in human skin, in the prostate, in the lungs and in the cornea [19]. The enzyme was cloned from hair routes and has been implicated in hair growth and skin development [19]. Later experiments indicated that the enzyme is also expressed in human monocytes when these cells are differentiated in vitro to macrophages. Stimulation of these cells with the Th2 cytokines interleukins-4 and -13 upregulated the expression of the enzyme [20]. When human peripheral monocytes are long-term stimulated with toll-like receptor-2/4 agonists, ALOX15B expression is also upregulated and this treatment induced a switch in the lipid mediator profile produced by these cells [21]. Despite these findings the biological relevance of ALOX15B orthologs is far from clear. A recent review nicely summarizes the potential biological functions of human ALOX15B, but this paper also indicates that there is no unifying concept for the biological role of ALOX15B [22].

Mouse Alox15b was first described in 1991 in phorbol ester-treated mouse skin [23]. This enzyme, which was later on cloned and expressed as recombinant protein [24] converted exogenous arachidonic acid to 8-HETE and it was therefore originally called mouse 8-LOX [23]. Today we know that this enzyme constitutes the mouse ortholog of human ALOX15B and it should therefore be called Alox15b. Mouse and human ALOX15B share a high (78%) degree of amino acid identity and the two *ALOX15B* genes have similar structures [1]. They both span some 10 Mbp and consist of 14 exons and 13 introns. Thus, from a structural point of view, the two enzymes are very similar. On the other hand, the two enzymes exhibit different catalytic properties, and these data suggest that their biological functions in mice and men might be different. One of the most striking differences between the two ALOX15B orthologs is their different product pattern when arachidonic acid is sued as the substrate. Human ALOX15B [19,25] catalyzes the stereospecific oxygenation of AA to 15S-HpETE (15S-hydroperoxy-5Z,8Z,11Z,13E-eicosatetraenoic acid). In contrast, mouse Alox15b [23] converts the same substrate to 8S-HpETE (8S-hydroperoxy-5Z,9,9E,11Z,14Z-eicosatetraenoid acid). Interestingly, such differences in the product mixture of the two enzymes were not observed when complex substrates such as AA-containing phospholipids and/or AA containing phospholipid loaded nanodiscs were used as the substrate [11]. In fact, with these substrates 15-HETE was identified as the major oxygenation product for mouse and human ALOX15B orthologs [11].

The molecular basis for the differential product patterns of mouse and human ALOX15B has been explored and in vitro mutagenesis studies indicated that the Tyr603Asp+His604Val mutant of mouse Alox15b formed similar reaction products as wildtype human ALOX15B [25]. An inverse mutagenesis strategy (Asp602Tyr+Val603His exchange in human ALOX15B) partially murinized the product pattern of the human enzyme [25]. To test whether the product pattern of mouse Alox15b can also be humanized in vivo, knock-in mice were recently created, which express the humanized AA 15-lipoxygenting Tyr603Asp+His604Val double mutant of mouse Alox15b instead of the AA 8-lipoxygenating mouse wildtype enzyme [26]. These mice are viable and breed normally [26], but the erythrocyte parameters of male knock-in individuals are compromised [26].

As indicated above, the Asp602Tyr+Val603His exchange murinized the product pattern of human ALOX15B with arachidonic acid and an inverse mutagenesis strategy on mouse Alox15b (Tyr603Asp+His604Val exchange) humanized the product mixture of this enzyme. However, whether this humanization of mouse Alox15b and murinization of human ALOX15B can also be observed when other polyenoic fatty acid are used as substrate has not been explored in detail. To fill this gap, we expressed the four enzyme variants as recombinant proteins and tested under strictly comparable experimental conditions their reaction specificities with different substrates (oxygenated and non-oxygenated polyenoic fatty acids). We found that human and mouse ALOX15B orthologs exhibit differential reaction specificities with all free polyenoic fatty acid derivatives tested (AA, EPA, DHA, LA, 15-HETE, and 8-HETE) and that the product pattern of the mouse enzyme can be humanized by Tyr603Asp+His604Val exchange. When an inverse mutagenesis strategy was applied for human ALOX15B, the product mixture was murinized for AA and EPA oxygenation, but the major DHA oxygenation product was identified as 7-HDHA. Thus, with this substrate fatty acid the product pattern was not murinized. To explore the molecular basis for the observed functional differences we carried out substrate docking studies and performed molecular dynamics simulations, and the obtained in silico data are consistent with our experimental observations.

## 2. Results

### 2.1. Recombinant Expression of Mouse and Human ALOX15B Orthologs and of Selected Mutants

Human ALOX15B oxidizes arachidonic acid to 15-HETE but its D602Y+V603H mutant forms a 2:1 mixture of 8-HETE and 12-HETE [25]. Thus, with arachidonic acid the D602Y+V603H double mutant of human ALOX15B forms similar products as wildtype mouse Alox15b. On the other hand, mouse Alox15b forms almost exclusively to 8-HETE from AA but its Y603D+H604V mutant dominantly produces 15-HETE [25] and thus, this enzyme mutant forms humanized reaction products [25].

To test whether murinization of human ALOX15B and humanization of mouse Alox15b ortholog can also be seen when other substrate fatty acids are used, we expressed the four enzyme species (wildtype human ALOX15B, D602Y+V603H mutant of human ALOX15B, wildtype mouse Alox15b, and Y603D+H604V mutant of mouse Alox15b) as his-tag fusion proteins in *E. coli* and quantified the expression levels of the four enzyme variants by quantitative immunoblotting using an anti-his-tag antibody (Figure 1). The expression levels of the different recombinant proteins are summarized in (Table 1). It can be seen that all ALOX15B variants are well expressed in *E. coli* and that mouse Alox15b reached an expression level of about 700 mg Alox15b protein per liter bacterial liquid culture. Next, we attempted to purify the recombinant proteins by affinity chromatography on Ni-agarose gel. The two wildtype enzymes could well be purified as catalytically active proteins employing this purification technique.

Unfortunately, the D602Y+V603H mutant of human ALOX15B lost more than 90% of its catalytic activity during the purification procedure. To ensure strict comparability of the functional data we used crude enzyme preparations (bacterial lysate supernatant) for functional characterization.

### 2.2. Mouse and Human ALOX15B Variants Prefer C20 and C22 Polyenoic Fatty Acids over Linoleic Acid (C18)

To test the substrate specificity of human and mouse ALOX15B variants in vitro activity assays using linoleic acid (LA), 5,8,11,14,17-eicosapentaenoic acid (EPA), arachidonic acid (AA), and 4,7,10,13,16,19-docosahexaenoic acid (DHA) as ALOX substrates were performed. When we set the catalytic activity of the different enzyme variants with AA 100% and calculated the relative catalytic activities with the other polyenoic fatty acids we found (Figure 2A, left columns) that wildtype human ALOX15 prefers DHA as substrate. EPA and AA were also oxygenated with relatively high reaction rates, but linoleic acid was not a good substrate for this enzyme. This data is consistent with previous reports on the substrate specificity of human ALOX15B [19].

When we tested the substrate specificity of the human ALOX15B D602Y+V603H double mutant DHA, EPA and AA were equally well oxygenated but LA was not a good substrate (Figure 2A, right columns). Similar experiments were subsequently carried out with mouse Alox15b and its Y603D+H604V double mutant. The wildtype enzyme converted DHA, EPA, and AA with similar reaction rates but here again, LA was not a good substrate (Figure 2B, left columns). For the mouse Alox15b Y603D+H604V double mutant EPA was the preferred polyenoic fatty acid substrate followed by DHA and AA. For this enzyme variant, LA was also not a good substrate in the absence of detergents (Figure 2B, right columns). The addition of sodium cholate improved the substrate behavior of LA [28] and this effect might be related to the allosteric properties of this enzyme [29]. However, since cholate is absent in most mammalian cells at these relatively high concentrations, we did not test the impact of this detergent on the reaction rate and on the product pattern in this study.

It should be stressed at this point that the data shown in Figure 2 characterize the product mixtures formed by the four different recombinant enzymes during a 5 min incubation period (end point measurements). Unfortunately, more detailed kinetic studies could not be carried out since our crude enzyme preparation did not allow spectrophotometric measurements following the increase in absorbance at 235 nm (formation of conjugated dienes).

### 2.3. D602Y+V603H Exchange Partly Murinized the Reaction Products Formed by Human ALOX15B with EPA and AA but Altered the Product Pattern with DHA in a Different Way

Human ALOX15B converts AA to 15-H(p)ETE [19,25] but D602Y+V603H exchange partly murinized the product pattern [25]. In fact, for this double mutant 8S-H(p)ETE was identified as major AA oxidation product. Whether such murinization of the product pattern can also be observed with other substrates has not been explored. To answer this question, we incubated wildtype human ALOX15B and its D602Y+V603H double mutant with different polyenoic fatty acids and analyzed the reaction products by RP-HPLC. As shown in Figure 3A, 15-H(p)ETE was the almost exclusive AA oxygenation product formed by the wildtype enzyme. However, for the D602Y+V603H double mutant a mixture of 8-H(p)ETE, 15-H(p)ETE, and 5-H(P)ETE was analyzed. This data confirmed the previous finding that D602Y+V603H exchange partly murinized the product mixture of human ALOX15B [25] but our results also indicate that the mutant enzyme formed a significant share of 5-HETE as minor (15–20% of the total conjugated dienes formed) side product. Interestingly, combined NP/CP-HPLC indicated that this minor side product was completely chiral (5S-HETE), suggesting that the 5S-HETE formation was completely enzyme-controlled.

Similar incubations were performed with EPA (Figure 3B) and DHA (Figure 3C). As expected from the product profile of AA oxygenation, 15-HEPE was the major EPA oxygenation product formed by wildtype human ALOX15B. In contrast, 8-HEPE was the major EPA oxygenation product formed by the D602Y+V603H double mutant of human ALOX15B. In addition, smaller amounts of 15-H(p)EPE and 5-H(p)EPE were found. Thus, the oxygenation products formed from AA and EPA by wildtype human ALOX15B and its D602Y+V603H double mutant were very similar. In contrast, with DHA the situation was somewhat different. Wildtype human ALOX15B oxygenated this highly unsaturated polyenoic fatty acid almost exclusively to 17-HDHA and this simple product pattern could be predicted on the basis of the analysis of the AA and EPA oxygenation products. Interestingly, the major DHA oxidation product formed by the D602Y+V603H double mutant was not 10-HDHA (the equivalent DHA oxygenation product of 8-HETE) but 7-HDHA instead. This product is the equivalent DHA oxygenation product of 5-HETE. 17-HDHA and 10-HDHA were also detected in minor quantities.

### 2.4. Y603D+H604V Mutation Humanized the Product Pattern of Mouse Alox15b with AA, EPA and DHA

Mouse Alox15b converts AA to 8-HETE but the Y603D+H604V exchange humanized the product mixture of this enzyme [25]. For this mutant, 15S-HETE was identified as the major AA oxidation product [25]. When we incubated mouse Alox15b and its Y603D+H604V mutant with AA we confirmed these results. 8-HETE was the major wildtype enzyme product whereas the Y603D+H604V double mutant produced 15-HETE (Figure 4A). When EPA was used as the substrate, 8-HEPE was analyzed as the major oxygenation product of the wildtype enzyme, but 15-HEPE was dominantly formed by the Y603D+H604V double mutant (Figure 4B).

Finally, we explored the DHA oxygenation products formed by these two enzyme variants and found that, as expected from the product profiles of AA and EPA oxygenation, 10-HDHA was mainly formed by the wildtype enzyme whereas 17-HDHA dominated the product pattern produced by the Y603D+H604V double mutant (Figure 4C). Taken together these data indicated that the Y603D+H604V exchange humanized the product pattern of mouse Alox15b not only with AA but also with EPA and DHA. The outcome of the DHA experiments was of particular interest since the inverse mutagenesis strategy of human ALOX15B (D602Y+V603H exchange) murinized the reaction products of AA and EPA oxygenation by human ALOX15 (Figure 3A,B) but not the product pattern of DHA oxygenation (Figure 3C).

### 2.5. D602Y+V603H Exchange in Human ALOX15B and Y603D+H604V Mutation of Mouse Alox15b also Altered the Reaction Products of Linoleic Acid Oxygenation

In the absence of detergents, linoleic acid was not a good substrate for either human or mouse ALOX15B (Figure 2). Nevertheless, we analyzed the product mixture formed by the four different enzyme variants from these polyenoic fatty acids, which abundantly occurs in most mammalian cells. For this purpose, in vitro activity assays were carried out (Figure 2), and the conjugated dienes formed were prepared by RP-HPLC and further analyzed by the combined normal phase/chiral phase HPLC (NP/CP-HPLC). With linoleic acid as the substrate only four major ALOX products (13*S*-HODE, 13*R*-HODE, 9*S*-HODE, 9*R*-HODE) can be formed and all of them are well separated by our combined NP/CP-HPLC analytical protocol.

As expected, the major LA oxygenation product of wildtype human ALOX15B was 13*S*-HODE. 13*R*-HODE and the two 9-HODE enantiomers (9*S*- and 9*R*-HODE) were only formed in small amounts (Figure 5A). Interestingly, the major LA oxygenation product formed by the mouse wildtype Alox15b was 9S-HODE and here again, the other HODE isomers were only minor side products (Figure 5C). The human ALOX15B D602Y+V603H double mutant exhibited only a very low catalytic activity with this fatty acid and here we observed a complex mixture of all four LA oxygenation products. However, 9- and 13-HODE were not complete racemic mixtures since for each positional isomer the S-enantiomer prevailed (*S/R*-ratio of 7:3) for the two positional isomers. The Y603D+H604V double mutant of mouse Alox15b dominantly formed 13*S*-HODE and thus, the product pattern of the mouse Alox15b with linoleic acid was fully humanized.

### 2.6. MD Simulations Suggest an Alternative Fatty Acid Binding at the Active Site of Human and Mouse ALOX15 Orthologs

Human and mouse ALOX15 orthologs share a high degree of amino acid conservation but the catalytic properties of the two enzymes are remarkably different. Human ALOX15B oxygenates AA and EPA to the 15-oxygenation products (Figure 3A,B) whereas mouse Alox15b dominantly forms the 8-oxygenation products from these two PUFAs. A plausible explanation for the different product patterns was an inverse substrate alignment at the active site of the two enzymes. For 15-HETE formation by human ALOX15B a tail-first substrate alignment at the active site can be suggested [30]. Such substrate binding allows hydrogen abstraction from C13 and [+2] radical rearrangement. In contrast, 8-lipoxygenation by mouse Alox15b may involve head-first substrate alignment, hydrogen abstraction from C10, and [−2] radical rearrangement.

To test these hypotheses, we have carried out docking calculations of the substrates AA, EPA, DHA, and LA at the active sites of the four enzyme variants. For each case, we have selected the five more stable structures and classified them according to the preference of tail-first (TF) or head-first (HF) substrate alignment. The results are shown in Table 2. Despite the limited accuracy of such docking calculations, it can be seen that the four substrates preferentially tend to adopt a tail-first alignment in wildtype human ALOX15B and in the Y603D+H604V mutant of mouse Alox15b. In contrast, a head-first orientation was preferred in the D602Y+V603H mutant of human ALOX15B and in wildtype mouse Alox15b. To visualize these results, images of the tail-first alignment of AA (the carboxylate pointing to the protein surface, Figure 6A) in wildtype human ALOX15B, and of the head-first alignment of AA in wildtype mouse Alox15b (the carboxylate facing to the bottom of the enzyme, Figure 6B) are illustrated. In both cases, AA adopts a U-shaped structure in the substrate binding pocket.

After our docking calculations revealed the preferred substrate alignment for each enzyme-substrate complex, we employed molecular dynamics (MD) simulations to explore which hydrogen atoms are located at a suitable position for productive hydrogen abstraction from a bisallylic methylene. In principle, AA involves three different bisallylic methylene groups and each of these carbon atoms carry two hydrogen atoms, which can be abstracted during initial hydrogen removal. To explore which of these hydrogen atoms are suitable candidates for hydrogen abstraction, we focused on AA and ran eight MD simulations (100 ns each). These simulations mirror both HF and TF substrate orientation at the active site of the four different enzyme species (wildtype human ALOX15B, D602Y+V603H mutant of human ALOX15B, wildtype mouse Alox15b, and Y603D+H604V mutant of mouse Alox15b). Each simulation started from the energetically most favorable complex obtained in our docking studies. The evolution of the distances between the oxygen atom of the Fe(III)-OH^−^ cofactor (catalytically competent ALOX species) and the closest hydrogen atom attached to C7 (H7), C10 (H10) and C13 (H13) of AA along the MD simulations is pictured in Figure 7 and Appendix A and the corresponding average distances are summarized in Table 3.

In wildtype human ALOX15B the carboxylate group of AA in the HF alignment is destabilized by D602 and V603. A similar destabilizing effect was observed by D603 and V604 in the humanized Y603D+H604V double mutant of mouse Alox15b. Because of these destabilizing effects, AA is unable to establish a strong non-covalent interaction with any residue in the depth of the enzyme cavity and it tends to move away towards the protein surface. The large average distances given in Table 3 confirm this trend.

Conversely, the strong interaction between the AA carboxylate and Arg428 stabilizes the position of TF AA alignment inside the substrate binding pocket. Thus, the tail-first orientation turns out to be the preferred way of substrate alignment and this data is consistent with the docking calculations and with the observed 15-HETE formation by wildtype human ALOX15B and the humanized mouse mutant. In our MD simulations many structures with H13 and/or H10 at distances suitable for hydrogen abstraction appear for both wildtype human ALOX15B, and the humanized Y603D+H604V double mutant of mouse Alox15b. Consequently, the average distances varying between 3.2 and 4.8 Å are rather small.

For those enzyme variants carrying a Tyr and a His in the core of the enzymes (wildtype mouse Alox15b and D602Y+V603H mutant of human ALOX15B) the AA carboxylate can form stabilizing hydrogen bonds independently of whether AA is aligned with HF or TF. In both cases, the enzyme-substrate complex remains stable, with H13 and H10 being located close to the oxygen of the cofactor (Figure 7 and Appendix A, Table 3).

In any alignment for any of the four studied enzymes H7 was always rather distant from Fe(III)-OH^−^ cofactor and thus hydrogen abstraction from this pro-chiral center is unlikely. However, our MD simulations cannot decide if the HF or the TF substrate alignment is more favorable, but our docking results already suggested that HF substrate orientation is the preferred choice for mouse Alox15b and the murinized human ALOX15B double mutant. The potential energy barriers for hydrogen abstractions from the pro-chiral bisallylic methylenes of AA (H7, H10, H13) tend to increase with the distance between the hydrogen to be abstracted and the oxygen atom of the Fe(III)-OH^−^ cofactor. However, there is no precise correlation between the energy barriers and the distances since many other factors are involved. AA 15-lipoxygenation by wildtype human ALOX15B and by the mouse Alox15b Tyr603Asp+His604Val double mutant involves hydrogen abstraction from C13 (H13). The nascent pentadienyl radical formed during hydrogen abstraction is close to planar (sp2 hybridization of the carbon atoms) and the π-electron density is completely delocalized over the entire pentadienoic system (C11 to C15). In contrast, AA 8-lipoxygenation by wildtype mouse Alox15b and by the human ALOX15B Asp602Tyr+Val603His double mutant involves hydrogen abstraction from C10 (H10). Here the π-electron density of the pentadienyl radical is delocalized over C8 to C12

We have previously suggested that the extent of conversion of the nonplanar structure of the fatty acid substrate to the planar structure of the evolving pentadienyl radical (geometric motion) severely impacts the energy barrier of the hydrogen abstraction. In fact, a high degree of geometric motion appears to increase the energy barrier [31]. The differential steric hindrance of this geometric motion by the C-terminal amino acids (Ile675 for human ALOX15B and Ile677 for mouse Alox15) as well as by Leu609 (for human ALOX15B) and Leu611 (for mouse Alox15b) appears to favor C13 hydrogen abstraction by human ALOX15B but C10 hydrogen abstraction by its mouse ortholog. For TF binding of AA in wildtype human ALOX15B (Figure 8) the dihedral angles C11-C12-C13-C14 and C12-C13-C14-C15 are closer to planarity (0° or 180°) than the dihedral angles C8-C9-C10-C11 and C9-C10-C11-C12. Thus, abstraction of H10 requires a high degree of geometric motion of AA in the region of the C8-C9 double bond but the side chains of Ile675 and Leu609 significantly hinder this motion. Thus, H13 abstraction and subsequent [+2] radical rearrangement is sterically preferred. For tail-first AA alignment at the active site of the Y603D+H604V double mutant of mouse Alox15b we observed a similar steric situation (Appendix A).

On the other hand, the scenario is the opposite for head-first AA alignment. For wildtype mouse Alox15b (Appendix A) and for the D602Y+V603H double mutant of human ALOX15B (Appendix A) the steric effects of the above-mentioned amino acids favor the H10 abstraction and subsequent [−2] radical rearrangement.

### 2.7. D602Y+V603H Exchange in Human ALOX15B and Y603D+H604V Mutation in the Mouse Ortholog also Altered the Substrate Behavior of HETE Isomer

All canonic HETE-isomers involve bisallylic methylenes and thus they constitute suitable ALOX substrates. In theory, 15-HETE should be a good substrate for mouse Alox15b, and 8,15-diHETE is predicted as major reaction product. On the other hand, 8-HETE should not be accepted as substrate by wildtype mouse Alox15b. In contrast, human ALOX15B should well oxygenate 8-HETE and here again, 8,15-diHETE should be the major reaction product. However, 15-HETE should not be oxygenated by wildtype human ALOX15B. To test these hypotheses, in vitro activity assays were carried out with wildtype human and wildtype mouse ALOX15 orthologs and the reaction products were analyzed by RP-HPLC.

As indicated in Figure 9, wildtype human ALOX15B accepted 8-HETE but not 15-HETE as the substrate. As expected, mouse Alox15b exhibited an inverse substrate specificity. 15-HETE was a good substrate, but 8-HETE was not (Figure 9). D602Y+V603H exchange murinized the substrate specificity of the human ALOX15B since 15-HETE was a good substrate for this enzyme variant. In contrast, Y603D+H604V mutation in mouse Alox15b humanized the substrate specificity of mouse Alox15b since for this enzyme mutant 8-HETE was a good oxygenation substrate whereas 15-HETE was not oxygenated (Figure 9).

### 2.8. Wildtype Mouse Alox15b Forms Significant Amounts of 8,15-diHETE from AA during Long Term Incubations

When we analyzed the mixture of oxygenation products formed from AA during short-term incubations by mouse Alox15b, we only detected the formation of 8-HETE and 15-HETE. However, after longer incubation periods we observed the additional formation of conjugated trienes that co-migrated with an authentic standard of 8S,15S-diHETE (Figure 10). This compound, which was absent in the product mixture of short-term incubations, involved the canonic conjugated triene chromophore (inset to Figure 10) and this data is consistent with its chemical structure.

In principle, there are two distinct biosynthetic pathways for 8,15-diHETE formation by mouse Alox15b. The first mechanism involves primary formation of 8-HETE, which can subsequently be used as substrate for secondary 15-lipoxygenation. The second pathway involves primary AA 15-lipoxygenation followed by 8-lipoxygenation.

### 2.9. The Product Pattern of AA Oxygenation by Mouse Alox15b Depends on the Duration of the In Vitro Incubation Period and on the Amounts of Enzyme Added

After a 3 min incubation period of mouse Alox15b with AA, a 93:7 mixture of 8-HETE and 15-HETE was analyzed (Figure 4A). Next, we explored whether the product pattern formed by mouse Alox15b depends on the enzyme concentration. For this purpose, we first incubated low amounts of enzymes for different time periods and analyzed the product pattern (Figure 11A). Here we found that at all time points 8-HETE was the dominant reaction product (black symbols) whereas 15-HETE as well as 8,15-diHETE were only formed in small amounts. However, at higher enzyme concentrations this product pattern was remarkably different (Figure 11B). When activity assays were carried out at a 10-fold higher enzyme concentration the relative share of 8-HETE formation was down, whereas 8,15-diHETE formation was augmented. In fact, at this enzyme concentrations 8,15-diHETE was the dominant AA oxygenation product (Figure 11B).

### 2.10. MD Simulations of 15-HETE and 8-HETE Binding

To explore the structural basis for differential oxygenation of 15- and 8-HETE by human and mouse ALOX15B orthologs, we ran four independent MD simulations (100 ns each). For wildtype human ALOX15B and for the humanized mouse Alox15b Y603D+H604V double mutant we analyzed TF alignment of 8-HETE. Since 15-HETE was not a good substrate for these two enzymes (Figure 9) we did not analyze the binding of this substrate. For wildtype mouse Alox15b and the murinized human ALOX15B D602Y+V603H double mutant, 15-HETE (HF) was the preferred substrate (Figure 9) and thus, we did not initially explore 8-HETE binding.

The evolution of the distances between the oxygen atom in the Fe(III)-OH^−^ cofactor to the closest hydrogen atom attached to C7 (H7), C10 (H10) or C13 (H13) along the MD simulations is pictured in Figure 12, and the average distances are summarized in Table 4. It can be seen that H13 in 8*S*-HETE (TF) is localized in close proximity to the Fe(III)-OH^−^ cofactor and thus, it is ready to be abstracted by the two enzymes. These steric configurations are consistent with the formation of 8S,15S-diHETE from the 8S-HETE substrate. On the other hand, H10 in 15*S*-HETE (TF) is also localized in close proximity to the Fe(III)-OH^−^ cofactor and thus, it may also be ready to be abstracted by both wildtype mouse Alox15b and the murinized D602Y+V603H double mutant of human ALOX15B. These conformations of the enzyme-substrate complexes explain the formation of 8S,15S-diHETE from 15-HETE by these two enzymes, which was experimentally observed.

Another aspect needs to be considered when the formation 8S,15S-diHETE from AA by wildtype mouse Alox15b is explored. We have shown that the primary formation of 8S-HETE is a favorable and fast process. In contrast, the formation of 8S,15S-diHETE from 8S-HETE is an unfavorable and slow process, but it is in fact possible. In effect, according to Table 2, the tail-first alignment of AA in wildtype mouse Alox15b is not the most favorable one, but it is possible. So, the tail-first alignment of 8S-HETE is also possible, and the corresponding MD simulation (Table 4 and Appendix A) shows that if this is the case, H13 is ready to be abstracted to yield the secondary 15-lipoxygenation. Moreover, Figure 9 shows that 8S-HETE is not a good substrate of wildtype mouse Alox15b, but it is somewhat metabolized. A second pathway would involve an initial slow 15-lipoxygenaton (it would imply an unfavorable tail-first orientation of AA), but a fast secondary 8-lipoxygenation of 15-HETE. Both pathways seem to be possible, as both are globally slow, because 15-lipoxygenation is always unfavorable.

### 2.11. Impact of Reaction Conditions on the Product Pattern of Mouse and Human ALOX15B

For the soybean LOX1 it has previously been reported [32] that the pattern of the reaction products formed from linoleic acid depends on the pH of the reaction mixture. On the other hand, the product patterns of a number of vertebrate ALOX isoforms [33] were remarkably stable when the pH of the reaction mixture was altered.

To explore the possible impact of the reaction conditions on the rate of EPA oxygenation, we incubated human and mouse ALOX15B orthologs with EPA at different pH (6.4, 7.4, and 8.4), different temperatures (15 °C, 25 °C, and 35 °C) and different substrate concentrations (50 µM, 100 µM, and 200 µM) for 5 min, quantified the reaction products formed during the incubation period and determined the chemical structure of the major EPA oxygenation products. As indicated in Table 5 (upper panel), human ALOX15B converted EPA most rapidly at pH 7.4. At pH 6.4 and 8.4 the oxygenation rates were gradually reduced but at all pH-values 15-HEPE was the exclusive oxygenation product. As expected, for mouse Alox15b 8-HEPE was the dominant EPA oxygenation product and 15-HEPE was only produced in small quantities. Interestingly, in contrast to human ALOX15B, which exhibited a higher EPA oxygenation rate at pH 7.4, mouse Alox15b was more active at pH 6.4. (Table 5, upper panel). When we explored the temperature dependence of EPA oxygenation (Table 5, middle panel) we found that both human and mouse ALOX15B oxygenated EPA with highest rates at 37 °C. At all temperatures 15-HEPE was the exclusive EPA oxygenation product of the human enzyme, whereas 8-HEPE was dominant for mouse Alox15b.

Finally, we also tested the impact of substrate concentrations (Table 5, lower panel). Human and mouse ALOX15B orthologs exhibited the highest EPA oxygenase activities at 200 µM substrate concentration. Here again, the exclusive EPA oxygenation product was identified as 15-HEPE for human ALOX15B at all substrate concentrations tested. For mouse Alox15b the situation was somewhat different. At low substrate concentrations (50 µM) a 1:2 mixture of 15- and 8-HEPE was analyzed but the relative share of 8-HEPE did increase with the increasing substrate concentrations. In fact, at 200 µM EPA 8-HEPE was rather dominant (Table 5, lower panel).

## 3. Discussion

### 3.1. Degree of Novelty, Advancement of Knowledge, and Limitations

The product pattern of wildtype mouse and human ALOX15B orthologs with AA as the substrate has previously been explored and product analysis indicated that human ALOX15B converts AA mainly to 15S-HETE [19,25] whereas 8S-HETE was the dominant AA oxygenation product of mouse Alox15b [25]. Y603D+H604V exchange in mouse Alox15b humanized the product pattern with AA as the substrate and the inverse mutagenesis strategy on the mouse enzyme partly humanized this enzyme property [25]. For wildtype human ALOX15B the product pattern with EPA and DHA has also been explored [34], but a number of important questions remained unanswered: (i) Which products are formed from other PUFAs by mouse Alox15b. (ii) Does D602Y+V603H exchange in human ALOX15B murinize the product pattern of this enzyme when EPA, DHA, and LA are used as the substrate. (iii) Does Y603D+H604V exchange in mouse Alox15b humanize the product mixture of this enzyme when EPA, DHA, and LA are used as substrate. (iv) Do wildtype mouse and human ALOX15B orthologs and their humanized/murinized double mutants accept hydroxylated PUFAs (8-HETE and 15-HETE) as the substrate and what are the major products of these reactions. (v) Can the different reaction specificities of mouse and human ALOX15B orthologs and their humanized/murinized double mutants be explained by the different ways of substrate binding at the active site of the enzyme variants (docking studies and MD simulations).

To address these questions, we designed appropriate experiments and obtained the following results: (i) 8-HEPE and 10-HDHA were the dominant oxygenation products of mouse Alox15b with EPA and DHA, respectively (Figure 3). (ii) Mouse Alox15b (Figure 4) converted AA, EPA, and DHA to their n-12 oxygenation products (8-HETE, 8-HEPE, 10-HDHA) and the Y603D+H604V exchange humanized the product patterns with all three PUFAs. A similar humanization of the product mixture of mouse Alox15b was observed when linoleic acid was used as the substrate (Figure 5C,D). (iii) When an inverse mutagenesis strategy was employed for human ALOX15B, the D602Y+V603H exchange murinized the AA and EPA oxygenation products (Figure 4A,B). In contrast, when DHA was used as the substrate for the D602Y+V603H double mutant of human ALOX15B (Figure 4C) we did not see such murinization. Instead of 10-HDHA, which was expected to be formed by a murinized enzyme, we identified 7-HDHA as the major DHA oxygenation product. In other words, the D602Y+V603H double mutant of human ALOX15B converted DHA to the same reaction product that is formed from this substrate by ALOX5. This data suggests that the way of DHA binding at the active site of the D602Y+V603H double mutant of human ALOX15B should be similar in the way of substrate binding at ALOX5. When LA was used as the substrate, we neither observed murinization of the reaction products. In fact, the D602Y+V603H double mutant of human ALOX15B produced an unspecific product mixture of 13S-, 13R-, 9S-, and 9R-HODE and this data suggests that the oxygenation reaction with this fatty acid was not strictly controlled by the enzyme (Figure 5A,B). (iv) 15-HETE is a good substrate for wildtype mouse Alox15b and for the murinized human ALOX15B double mutant Y603D+H604V. In contrast, it is not well oxygenated by wildtype human ALOX15B and the humanized D602Y+V603H double mutant of mouse Alox15b. In contrast, 8-HETE is well oxygenated by wildtype human ALOX15B and the humanized mouse Alox15b double mutant Y603D+H604V. 8,15-diHETE was the major oxygenation product formed from both substrates. (v) Docking calculations indicate that the four polyenoic fatty acid substrates (AA, EPA, DHA, and LA) tested in this study preferentially tend to adopt a tail-first (TF) substrate alignment in wildtype human ALOX15B and in the humanized Y603D+H604V double mutant of mouse Alox15b. In contrast, a head-first (HF) substrate alignment was calculated for murinized D602Y+V603H mutant of human ALOX15B and wildtype mouse Alox15b. Molecular dynamics simulations using AA as the substrate revealed that binding of carboxylate group of the head-first oriented substrate in the substrate binding pocket would be destabilized by D602 and V603 in wildtype human ALOX15B, and by D603 and V604 in the humanized Y603D+H604V double mutant of mouse Alox15b. Thus, the substrate fatty acid is unable to establish a strong non-covalent interaction with any active site amino acid residue present at the bottom of the substrate binding pocket. Conversely, the strong interaction between the negatively charged carboxylic group of the substrate and the negatively charged side-chain of Arg428 stabilizes the TF substrate alignment. When Tyr and a His residues are present in the active site (wildtype mouse Alox15b and the murinized D602Y+V603H double mutant of human ALOX15B), the fatty acid carboxylate can form stabilizing hydrogen bonds with amino acid residues present either in the bottom region of the substrate binding cavity (HF substrate alignment) or with amino acids located close to the entrance of the substrate binding pocket (TF alignment). However, the docking results suggest that HF substrate orientation appears to be preferred. (vi) For each preferred substrate alignment both H13 (attached to C13) and H10 (attached to C10) of AA are located sufficiently close to the oxygen of the Fe(III)-OH^−^ cofactor for hydrogen abstraction, and steric hindrance by the N-terminal Ile675 and by the side-chain of Leu609 is important. For TF, AA alignment at the active site of wildtype human ALOX15B and of the humanized Y603D+H604V double mutant of mouse Alox15b the H13 abstraction is preferred. In contrast, for HF, AA alignment at the active site of wild type mouse Alox15B and of the murinized D602Y+V603H double mutant of human ALOX15B H10 abstraction is preferred.

The most serious limitation of this study is that functional characterization of the four enzyme variants was carried out with crude enzyme preparations. Although we were able to purify wildtype human and mouse ALOX15B as well as theY603D+H604V double mutant of mouse Alox15b by affinity chromatography on Ni-agarose as catalytically active proteins, we experienced a >90% loss in catalytic of the D602Y+V603H double mutant of human ALOX15B during the purification procedure. The mechanistic basis for this loss in catalytic activity has not been clarified but to ensure strict comparability of the functional data we decided to use crude enzyme preparations (bacterial lysate supernatant) for functional characterization. For human and mouse wildtype ALOX15B orthologs we carried out similar experiments with the purified enzyme preparations and obtained similar data as for the bacterial lysates.

### 3.2. Complex Ester Lipids as Substrate for Mouse and Human ALOX15B

Despite the differential product patterns with free AA human and mouse ALOX15, orthologs convert AA-containing phospholipids incorporated in nanodiscs [35] mainly to the 15-HETE derivatives [11]. Thus, the product mixture of the two enzymes with this substrate appears to be similar. We carried out similar studies using AA containing phospholipid liposomes and mitochondrial membranes as substrates. These preliminary experiments indicated that 15S-HETE containing phospholipids were the major oxygenation products formed by mouse and human ALOX15B and this data is consistent with the results obtained in the nanodisc experiment [11]. However, when we employed mitochondrial membranes or liposomes involving linoleic acid as substrate for the two enzymes, we observed different product patterns. For human ALOX15B, esterified 15S-HETE and 13S-HODE were the major oxygenation products, and the corresponding R-enantiomers were only formed in small amounts. When incubations were carried out with mouse Alox15b, esterified AA oxygenation products were only detected in small amounts. As the dominant PUFA oxygenation product we identified 9S-HODE. Thus, when mitochondrial membranes and LA-containing liposomes were used as substrate the product specificities of human and mouse ALOX15B are different. Whether similar differences in the reaction specificities of the two enzymes can be observed when other lipid-protein such as plasma membranes of different cells or lipoproteins are used as substrate must be tested experimentally in the future.

### 3.3. Impact of Reaction Conditions on the Product Pattern of ALOX15B Variants

The composition of the reaction products of most ALOX-isoforms with a given substrate is well defined and alterations in the reaction conditions in the near physiological range do not dramatically alter this enzyme property. There may be minor modifications but in principle the reaction specificities of ALOX-isoforms are very robust. This conclusion is supported by the experimental date shown in Table 5. On the other hand, we found that the pattern of the arachidonic acid oxygenation products strongly depends on the amount of enzyme added to the incubation mixture (Figure 11B). However, these data do not indicate that the specificity of the enzyme has changed. In fact, the differences in the product pattern are the consequence of enzyme kinetics. At high enzyme concentrations the fatty acid substrate (AA) becomes rate limiting and the primary reaction product accumulates. Under these conditions the primary oxygenation product is used as the substrate for secondary oxygenation (formation of 8,15-diHETE). Thus, also at high enzyme concentration the specificity of the enzyme largely remains unchanged.

There are, however, rare exceptions from this rule. For instance, the specificity of soybean LOX1 is more variable since the pattern of linoleic acid oxygenation products formed by this enzyme is pH-sensitive [32]. This result might be related to the alkaline pH_opt_ of this ALOX isoform. On the other hand, the specificity of AA oxygenation by most vertebrate ALOX-isoforms (human ALOX5, human ALOX12, human ALOX15, rabbit ALOX15, human ALOX15B, mouseAlox15, mouse Alox15b, zebrafishALOX12) was remarkable stable when the pH of the incubation mixture was varied in the near physiological range (pH 6.4–8.0) [33]. Here we observed (Table 5) that the product pattern of human ALOX15B was remarkably stable at different reaction conditions. At different pH-values (pH 6.4, 7.4, 8.4), at different temperatures (15 °C, 25 °C, 35°C), and at different substrate concentrations (50 µM, 100 µM, 200 µM) 15-HEPE was always identified as the exclusive EPA oxygenation product. In other words, in the near physiological range alterations in the reaction conditions did not impact the product pattern of human ALOX15B.

For mouse Alox15b the situation was somewhat different. For this enzyme we also observed that alterations in pH and temperature did hardly modify the product pattern of EPA oxygenation. Under all experimental conditions tested in our experiments (Table 5) 8-HEPE was identified as the dominant EPA oxygenation product (>90%) while 15-HEPE was only formed in smaller quantities (<10%). Interestingly, modification of the substrate concentration induced alterations in the reaction products of mouse Alox15b. At high substrate concentrations (200 µM) the relative share of the major EPA oxygenation product (8-HEPE) was about 95%. In contrast, at lower substrate concentrations we observed a gradual decline in the relative share of 8-HEPE formation. In fact, at 100 µM EPA 8-HEPE only contributed 85% to the product mixture and at 50 µM the 8-HEPE/15-HEPE ratio was only about 2:1. Thus, although at lower substrate concentrations 8-HEPE was still the major EPA oxygenation product, the relative share of 15-HEPE did significantly increase at lower substrate concentrations. The molecular basis for this observation was not explored but our findings need to be considered when the biosynthesis of oxygenated lipids in mouse cells, tissues, and body fluids (blood palsma, urine, liquor) are interpreted.

## 4. Materials and Methods

### 4.1. Chemicals

The chemicals used in this study were purchased from the following sources: Phosphate buffered saline without calcium and magnesium (PBS) from PAN Biotech (Aidenbach, Germany); nitrocellulose blotting membrane from Serva Electrophoresis GmbH (Heidelberg, Germany); EDTA (Merck KG, Darmstadt, Germany). Fatty acid substrates [5,8,11,14-eicosatetraenoic acid (AA), 5,8,11,14,17-eicosapentaenoic acid (EPA), 4,7,10,13,16,19-docosahexaenoic acid (DHA), 9,12-octadecadienoic acid (LA), 9,12,15-octadecatrienoic acid (ALA), 6,9,12-octadecatrienoic acid (GLA) 15S-HETE, 8S-HETE], and authentic HPLC standards of HETE-isomers used for HPLC analysis [15S-HETE, 15S/R-HETE, 12S/R-HETE, 12S-HETE, 8S/R-HETE, 5S-HETE, 8S,15S-diHETE] were obtained from Cayman Chem (distributed by Biomol GmbH, Hamburg, Germany). Acetic acid from Carl Roth GmbH (Karlsruhe, Germany); sodium borohydride from Life Technologies, Inc (Eggenstein, Germany); isopropyl-β-thiogalactopyranoside (IPTG) from Carl Roth GmbH (Karlsruhe, Germany); restriction enzymes from ThermoFisher (Schwerte, Germany); the E. coli strain Rosetta2 DE3 pLysS from Novagen (Merck-Millipore, Darmstadt, Germany). Oligonucleotide synthesis was performed at BioTez Berlin Buch GmbH (Berlin, Germany). Nucleic acid sequencing was carried out at Eurofins MWG Operon (Ebersberg, Germany). HPLC grade methanol, acetonitrile, n-hexane, 2-propanol, and water were from ThermoFisher (Schwerte, Germany).

### 4.2. Cloning and Expression of Mouse and Human ALOXB Orthologs

The cDNA sequences of mouse and human ALOX15B orthologs were extracted from the NCBI cDNA database and were chemically synthesized (BioCat GmbH, Heidelberg, Germany). For subcloning from the initial pUC57 synthesis vector into pET28b(+) (Novagen/Merck, Darmstadt, Germany) expression vector, a SalI restriction site was introduced immediately upstream of the start codon and a HindIII restriction site was generated immediately downstream of the stop codon. The sequences were optimized for bacterial expression by silent mutations.

The enzyme variants (wildtype and double mutants) were N-terminal hexa-his-tag fusion proteins, as described in Ref. [36]. Briefly, *E. coli* Rosetta2 (DE3)-pLysS cells were transformed with the recombinant plasmid pET28b and grown on an agar plate containing kanamycin and chloramphenicol. An isolated clone was selected and grown under shaking in a glucose free MSM with added trace elements and glucoamylase. Protein expression was induced by addition of IPTG. After 18 h, cells were harvested and homogenized by sonication, and the cell free supernatant was used as enzyme source.

### 4.3. Mutagenesis Studies

Site directed mutagenesis was performed as described in Ref. [36]. Plasmid DNA containing the coding sequence of the bony fish ALOX15 isoforms was incubated with specific primer pairs containing the required changes in their nucleotide sequence to achieve the intended amino acid exchanges and Pfu Ultra II Hot Start 2 × PCR Master Mix (Agilent Technologies, Waldbronn, Germany). After the PCR protocol (18 cycles), parent DNA was digested using *DpnI*. *E. coli* XL-1 Blue competent cells (Agilent Technologies Inc., Santa Clara, CA, USA) were transformed with the mutated plasmid and after replication the plasmid DNA of a selected bacterial colony was sequenced (Eurofins Genomics Germany GmbH, Ebersberg, Germany).

### 4.4. In Vitro Activity Assays and HPLC Analyses of the Reaction Products

Enzyme activity assays and analysis of the reaction products by HPLC were performed as described in Ref. [36]. Different amounts of the cell free supernatants were incubated with 100 µM substrate (AA, EPA, and DHA) and the reduced reaction products were analyzed by RP-HPLC on a Shimadzu instrument connected with a Hewlett Packard diode array detector 1040 A. Metabolites were separated on a Nucleodur C_18_ Gravity column (Macherey-Nagel, Düren, Germany; 250 × 4 mm, 5 μm particle size) coupled with a corresponding guard column (8 × 4 mm, 5 μm particle size). A solvent system consisting of acetonitrile:water:acetic acid (70:30:0.1, by volume for AA and DHA derivatives and 50:50:0.1, by volume for EPA derivatives) was used at a flow rate of 1 mL·min^−1^. For more detailed analysis of the reaction products of linoleic acid oxygenation the conjugated dienes formed during the incubation period were prepared by RP-HPLC and further analyzed by normal-phase HPLC (NP-HPLC) and/or chiral-phase HPLC (CP-HPLC). Normal-phase HPLC was performed using the solvent system n-hexane/2-propanol/acetic acid (100/2/0.1, by volume) on a Nucleosil 100-5 column (250 × 4.6 mm, 5 µm particle size). 12-HETE enantiomers were resolved on a Chiralpak AD-H column (Daicel Corp., Osaka, Japan) with a solvent system consisting of n-hexane/methanol/ethanol/acetic acid (96:3:1:0.1, by vol; 1 mL/min).

### 4.5. Miscellaneous Methods including Statistics

For SDS/PAGE, approximately 100 µg denatured protein of the bacterial lysate supernatants were analyzed on a 7.5% polyacrylamide gel. The separated protein bands were then transferred onto a Protran BA 85 Membrane (Carl Roth GmbH, Karlsruhe, Germany) and the blots were probed with an anti-his-tag-HRP conjugated antibody (Miltenyi Biotec GmbH, Bergisch Gladbach, Germany). Immunoreactive bands were visualized using the SERVALight Polaris CL HRP WB Substrate Kit (Serva Electrophoresis GmbH, Heidelberg, Germany). Chemiluminescence was detected on a FUJIFILM Luminescent Image Analyzer LAS-1000plus & Intelligent Dark Box II. For testing the pH-dependence of the product mixture of wildtype human and mouse ALOX15B orthologs, a 1:1 mixtures of 0.05 M sodium phosphate buffer and 0.05 M sodium borate buffer were used and the different pH values were adjusted at room temperature by the addition of 5 M NaOH or HCl, respectively. The protein concentrations in the bacterial lysates were quantified using Bradford Reagent for quantitative protein determination (AppliChem, VWR International GmbH, Darmstadt, Germany) according to the instructions of the vendor. For statistical calculations and figure design we used the GraphPad prism program (version 8.00, GraphPad Software, La Jolla, CA, USA).

### 4.6. Structural Modeling

Four ALOX15B 3D models were built. Human ALOX15B coordinates were obtained from the X-ray structure with PDB ID 4NRE [37] and the mouse Alox15b model was obtained from AlphaFold DataBase (UniProt ID O35936) [38,39]. The double mutant versions of both human ALOX15B and mouse Alox15b were manually constructed using the Rotamers module by UCSF Chimera. Protonation was carried out at pH 7.4 using the ProteinPrepare tool by PlayMolecule.org [40].

### 4.7. Docking Calculations

The GOLD5.8 program [41] was used for the docking calculations. ChemScore [42] was employed for scoring purposes.

### 4.8. MD Simulations

The best-scored solutions for each substrate-enzyme complex (enzyme complexed with AA in head-first (HF) and in tail-first (TF) conformations, 15S-HETE in head-first conformation and 8S-HETE in tail-first conformation) from docking calculations were used as the initial structures for MD simulations. Ff19SB has been used as a force field for standard residues, while Fe and its coordinates have been parametrized following the Seminario Method from MCPB [43]. Force constants and RESP charges have been obtained using the Gaussian16 program at the B3LYP/6-31G(d) level of theory. As recommended in Ref. [44], OPC waters [45] were chosen as the water model. An orthorhombic solvation box of pre-equilibrated water molecules was built with a buffer of 10 Å around the molecule and Na^+^ ions have been added for neutralization. The AMBER20’s CUDA (GPU) pmemd package was used for running the simulations. The initial structure was minimized, heated at a rate of 50 K from 0 to 300 K, and the system was then equilibrated for 20 ps after each step. A total of 1 ns of NPT ensemble at 300 K and 1 atm was run for stabilization of the water box density. Prior to production, an equilibration of the system using the NVT ensemble at 300 K during 10 ns was performed. After equilibration, 100 ns of production was calculated using the same NVT ensemble.

## 5. Conclusions

Functional experiments with different polyenoic fatty acids, molecular docking studies and molecular dynamics simulations of substrate binding at the active site of human and mouse ALOX15B orthologs and their murinized/humanized double mutants suggest that AA 15-lipoxygenation by human ALOX15 involves tail-first (TF) substrate alignment. In contrast, for AA 8-lipoxygenation by mouse Alox15b head-first (HF) substrate orientation is preferred. The Asp602Tyr+Val603His exchange murinized the product mixture of human ALOX15B with arachidonic acid and eicosapentaenoic acid but not with docosahexaenoic acid. An inverse mutagenesis strategy on mouse Alox15b (Tyr603Asp+His604Val exchange) did humanize the product pattern of arachidonic acid, eicosapentaenoic acid, and docosahexaenoic acid oxygenation.

## Figures and Tables

**Figure 1 ijms-24-10046-f001:**
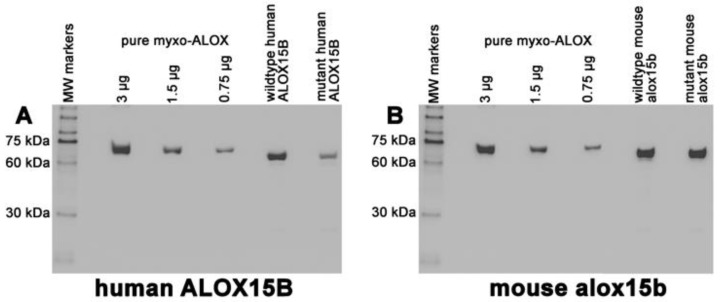
Quantification of the expression levels of human and mouse ALOX15B variants. ALOX15B variants were expressed in *E. coli* and aliquots of the bacterial lysis supernatants were analyzed by quantitative immunoblotting (see Section 4). A total of 1 µL human wildtype supernatant; 1.1 µL human D602Y+V603H mutant supernatant; 0.5 µL wildtype mouse supernatant; 1.2 µL mouse Y603D+H604V supernatant. The band intensity scale was calibrated applying known amounts of the pure recombinant N-terminal his-tag fusion protein of *M. fulvus* ALOX. (**A**) Human ALOX15B, (**B**) Mouse Alox15b.

**Figure 2 ijms-24-10046-f002:**
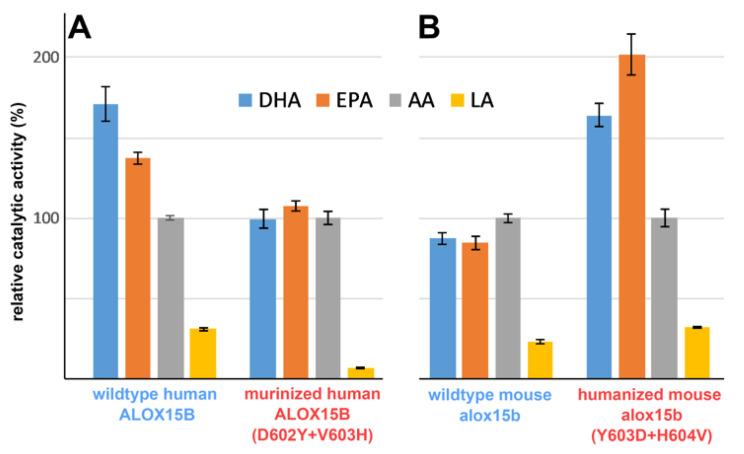
Substrate specificity of human and mouse ALOX15 variants with different polyunsaturated fatty acids. ALOX15B variants were expressed in *E. coli* and aliquots of the bacterial lysis supernatants were incubated in vitro for 5 min with different polyunsaturated fatty acids. After the incubation period the produced hydroperoxy fatty acids were reduced to their more stable hydroxy derivatives and products carrying a conjugated diene chromophore were quantified by RP-HPLC. Two incubation samples were set up for each fatty acid substrate and each sample was analyzed twice. The product formation from AA by each enzyme preparation was set 100% and the relative amounts of conjugated dienes formed from the other substrates were calculated. (**A**) Human ALOX15B, (**B**) mouse Alox15b.

**Figure 3 ijms-24-10046-f003:**
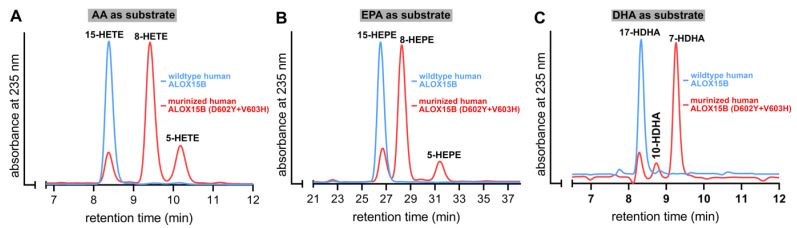
D602Y+V603H exchange partly murinized the product mixture of human ALOX15B with different polyenoic fatty acids. ALOX15B variants were expressed in *E. coli* and aliquots of the bacterial lysis supernatants were used to perform in vitro activity assays were carried out as described in the legend of Figure 2. Representative partial RP-HPLC chromatograms are shown. The chemical identity of the conjugated dienes formed during the 5 min incubation period was concluded by co-chromatography with authentic standards. Oxygenation products of AA and DHA were analyzed isocratically using a solvent system consisting of acetonitrile/water/acetic acid (70/30/0.1, by vol). The EPA oxygenation products were analyzed isocratically with the solvent system acetonitrile/water/acetic acid (50/50/0.1, by vol). (**A**) Arachidonic acid as substrate, (**B**) eicosapentaenoic acid as substrate, (**C**) docosahexaenoic acid as substrate.

**Figure 4 ijms-24-10046-f004:**
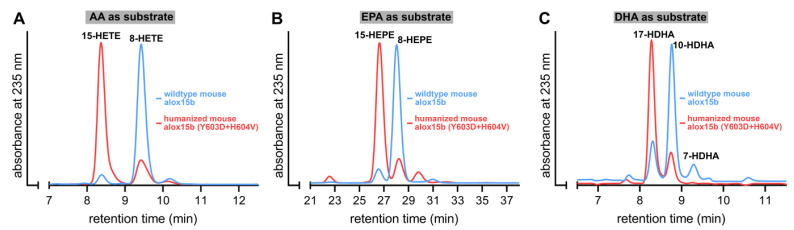
The Y603D+H604V exchange humanized the product pattern of mouse Alox15b with different polyenoic fatty acids. Mouse Alox15b variants were expressed in *E. coli* and aliquots of the bacterial lysis supernatants were used to perform in vitro activity assays as described in the legend of Figure 2. Representative partial RP-HPLC chromatograms are shown. The chemical identity of the conjugated dienes formed during the 5 min incubation period was concluded by co-chromatography with authentic standards. Oxygenation products of AA and DHA were analyzed isocratically using a solvent system consisting of acetonitrile/water/acetic acid (70/30/0.1, by vol). The EPA oxygenation products were analyzed isocratically with the solvent system acetonitrile/water/acetic acid (50/50/0.1, by vol). (**A**) Arachidonic acid as substrate, (**B**) eicosapentaenoic acid as substrate, (**C**) docosahexaenoic acid as substrate.

**Figure 5 ijms-24-10046-f005:**
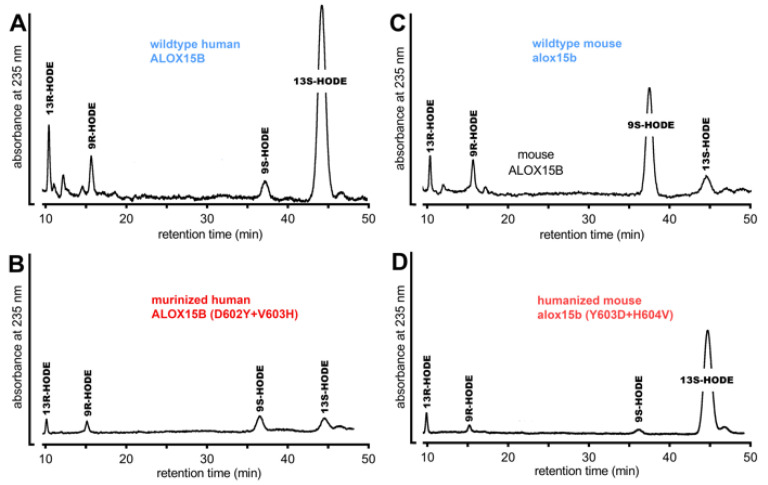
Product patterns of mouse and human wildtype ALOX15 orthologs and of their murinized (D602Y+V603H) and humanized (Y603D+H604V) double mutants using linoleic acid as the substrate. The different ALOX15B variants were expressed as N-terminal his-tag fusion proteins in *E. coli* and aliquots of the bacterial lysis supernatants were used to perform in vitro activity assays (Figure 2). The conjugated dienes formed during the incubation period were prepared by RP-HPLC and were further analyzed by NP/CP-HPLC. Representative partial RP-HPLC chromatograms are shown. The chemical identity of the conjugated dienes formed during the incubation period was concluded by co-chromatography with authentic standards. The chromatographic conditions of NP/CH-HPLC are described in detail in Section 4. (**A**) Wildtype human ALOX15B, (**B**) murinized human ALOX15B (D602Y+V603H), (**C**) wildtype mouse Alox15B, (**D**) humanized mouse Alox15b (Y603D+H604V).

**Figure 6 ijms-24-10046-f006:**
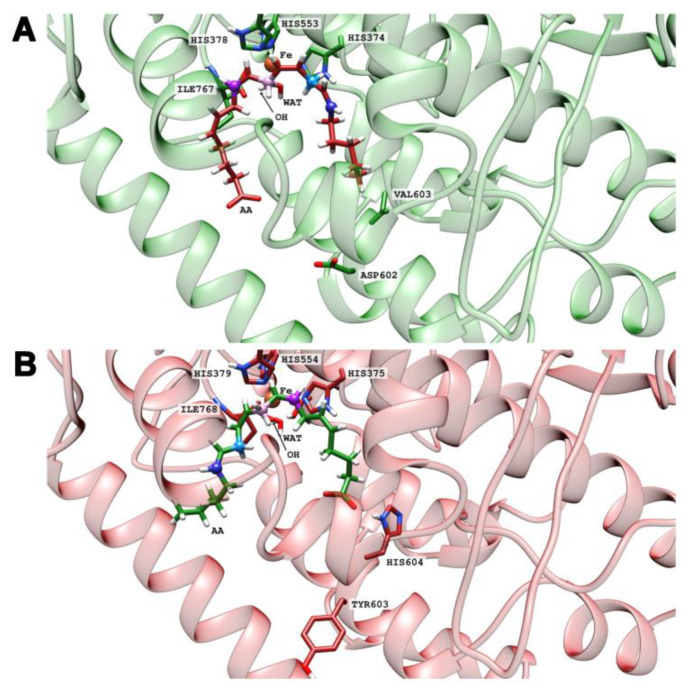
Tail-first alignment of AA (**A**) in wildtype human ALOX15B in green, and head-first alignment of AA (**B**) in wildtype mouse Alox15b in pink.

**Figure 7 ijms-24-10046-f007:**
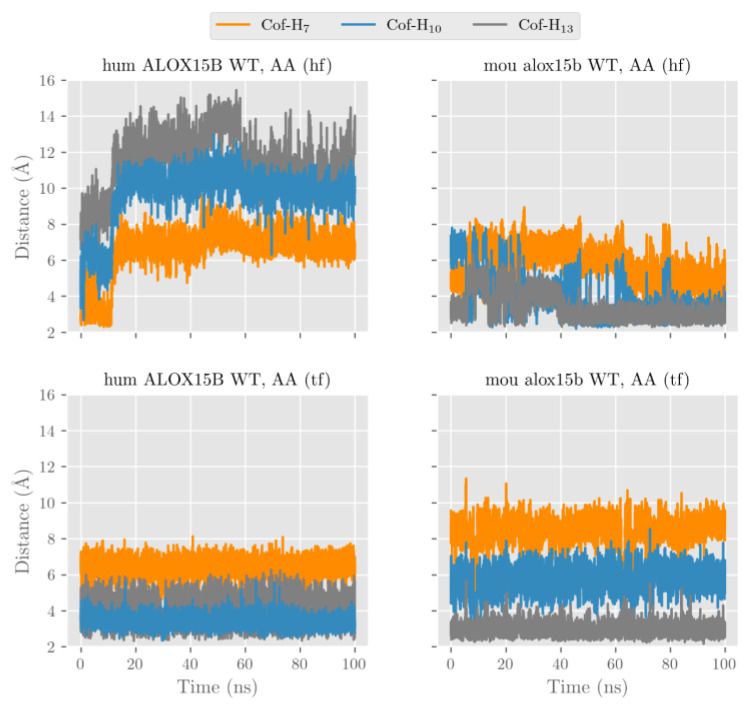
Distances from the oxygen atom in the Fe(III)-OH^−^ cofactor to the closest hydrogen atom attached to C7 (H7), C10 (H10), and C13 (H13) along the molecular dynamics simulations for AA (HF or TF) at the active site of wildtype human ALOX15B and wildtype mouse Alox15b.

**Figure 8 ijms-24-10046-f008:**
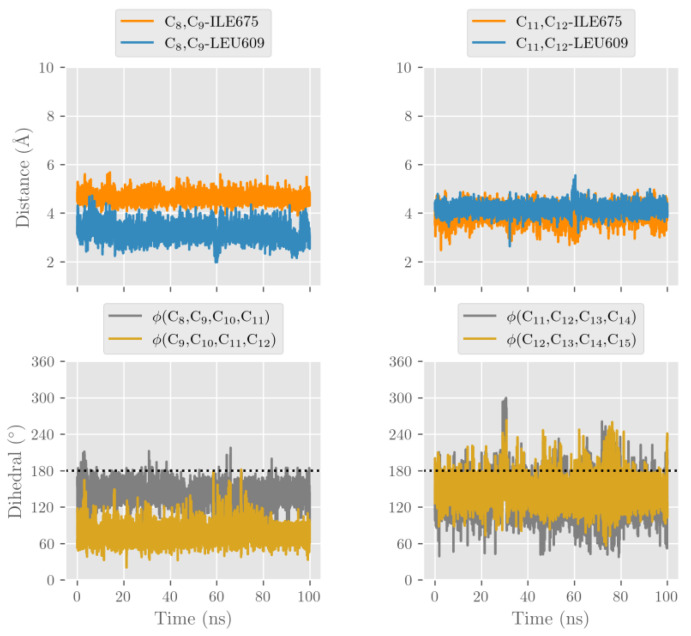
Shortest distances (Å) from the C8-C9 and C11-C12 double bonds to terminal Ile675 and Leu609, and dihedral angles C8-C9-C10-C11, C9-C10-C11-C12, C11-C12-C13-C14, and C12-C13-C14-C15 along the molecular dynamics simulation for AA (TF) at the active site of wildtype human ALOX15B.

**Figure 9 ijms-24-10046-f009:**
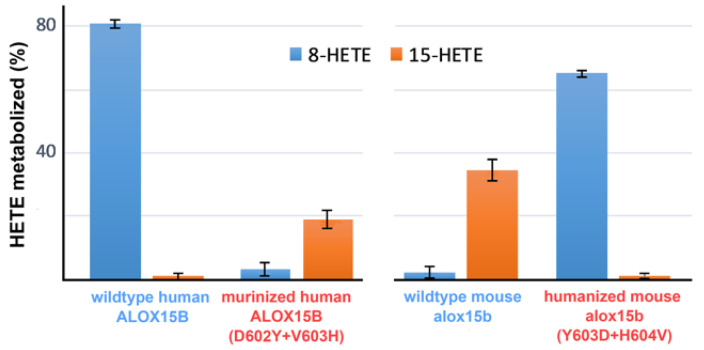
Substrate specificity of mouse and human wildtype ALOX15 orthologs and of their murinized (D602Y+V603H) and humanized (Y603D+H604V) double mutants using 15-HETE and 8-HETE as the substrate. The different ALOX15B variants were expressed as N-terminal his-tag fusion proteins in *E. coli* and aliquots of the bacterial lysis supernatants were used to perform in vitro activity assays using 15S-HETE and 8S-HETE (20 µM final concentration) as the substrate. The disappearance of the substrate during the incubation period was measured by RP-HPLC. Each activity assay was carried out in duplicate and each sample was analyzed twice by RP-HPLC.

**Figure 10 ijms-24-10046-f010:**
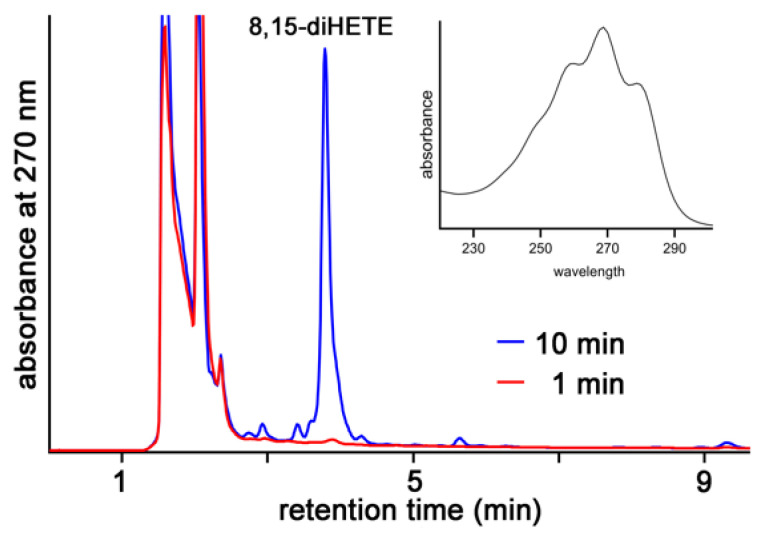
Formation of 8,15-diHETE during long-term incubations of AA with mouse Alox15b. Wildtype mouse Alox15b was expressed as N-terminal his-tag fusion protein in *E. coli* and aliquots of the bacterial lysis supernatant were used to perform in vitro AA oxygenation activity assays. The reaction was stopped after 1 (red) and 10 min (blue) by the addition of solid sodium borohydride and the reaction products were analyzed by RP-HPLC (legends to Figure 3 and Figure 4) recording the absorbance at 270 nm. Inset: UV-spectrum of the peak eluting at about 4 min indicating the presence of the canonic conjugated triene chromophore.

**Figure 11 ijms-24-10046-f011:**
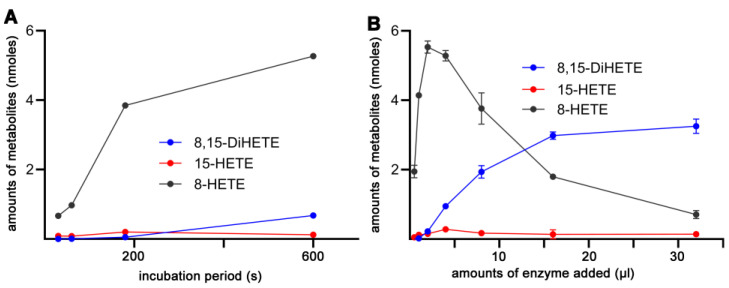
Formation of different oxygenation products by wildtype mouse Alox15b during the oxygenation of AA. Wildtype mouse Alox15b was expressed as N-terminal his-tag fusion protein in *E. coli* and aliquots of the bacterial lysis supernatant were used to perform in vitro AA oxygenation activity assays. The formation of 8-HETE, 15-HETE, and 8,15-diHETE (nmoles) was quantified. (**A**) Time dependence of product formation; single values are plotted. (**B**) Product formation at different enzyme concentrations. Each point represents the mean four different assays; means ± SD are given.

**Figure 12 ijms-24-10046-f012:**
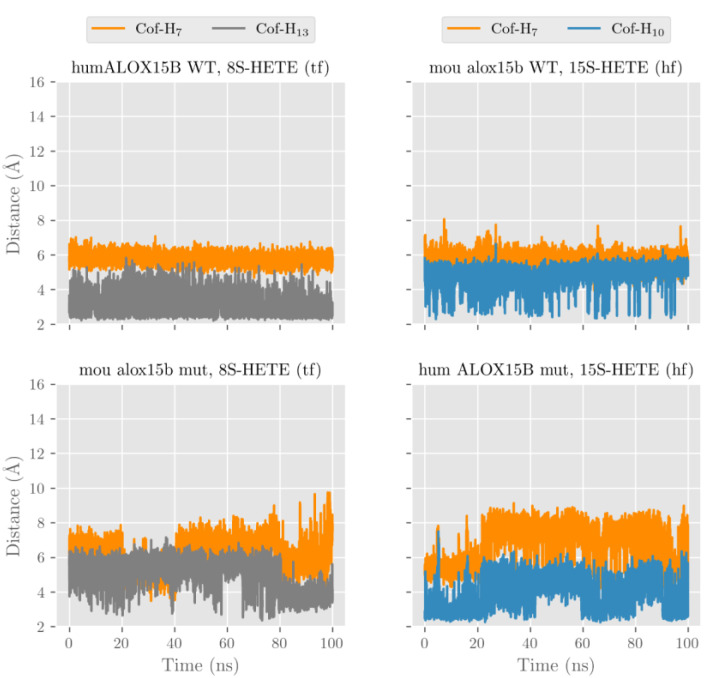
Distances from the oxygen atom in the Fe(III)-OH^−^ cofactor to the closest hydrogen atom attached to C7 (H7), C10 (H10), and C13 (H13) along the molecular dynamics simulations of tail-first 8*S*-HETE in wildtype human ALOX15B and Y603D+H604V mutant of mouse Alox15b, and head-first 15*S*-HETE in wildtype mouse Alox15b and D602Y+V603H mutant of human ALOX15B.

**Table 1 ijms-24-10046-t001:** Expression levels of the different recombinant ALOX15B variants. ALOX15B variants were expressed in *E. coli* and bacterial lysis supernatants were prepared as described in Section 4. Aliquots of the lysis supernatants were analyzed by quantitative immunoblotting using an anti-hist-tag antibody for specific staining of the ALOX bands. SDS-PAGE and immunoblotting were carried out as described in Section 4. The band intensity scale was calibrated by applying known amounts (3 µg, 1.5 µg, 0.75 µg) of the pure recombinant N-terminal his-tag fusion protein of *M. fulvus* ALOX as the reference protein [27].

Enzyme Variant	Expression Level (mg ALOX Protein per L Bacterial Liquid Culture)
Wildtype human ALOX15B	200
Wildtype mouse Alox15b	700
D602Y+V603H mutant (human ALOX15B)	170
Y603D+H604V mutant (mouse Alox15b)	292

**Table 2 ijms-24-10046-t002:** The five most stable structures of the substrates AA, EPA, DHA, and LA at the active sites of the four enzymes, according to docking calculations. Total free energy change (kJ/mol) that occurs on ligand binding and substrate alignment (Tail-first (TF) or Head-first (HF)). The TF solutions are highlighted in blue, and the HF solutions in pink.

Substrate	*AA*		*EPA*		*DHA*		*LA*	
Enzyme	ΔG^bind^	Orient.	ΔG^bind^	Orient.	ΔG^bind^	Orient.	ΔG^bind^	Orient.
**Human** **ALOX15B** **WT**	−57.26	TF	−64.38	TF	−69.23	TF	−58.15	TF
	−57.25	TF	−62.69	TF	−67.46	TF	−55.08	TF
	−57.21	TF	−59.66	TF	−62.39	TF	−54.38	TF
	−51.22	HF	−58.93	TF	−56.63	TF	−53.22	TF
	−51.00	TF	−56.36	TF	−55.85	HF	−52.51	TF
**Human** **ALOX15B mut**	−39.43	HF	−39.75	TF	−43.77	HF	−35.09	HF
	−39.13	HF	−39.34	HF	−43.71	TF	−34.82	TF
	−39.01	HF	−38.98	HF	−43.56	TF	−34.78	TF
	−38.97	HF	−38.93	TF	−43.51	HF	−34.70	TF
	−38.87	TF	−38.89	TF	−43.44	TF	−34.68	TF
**Mouse** **Alox15b WT**	−41.44	HF	−41.47	HF	−45.98	HF	−36.15	HF
	−40.12	TF	−41.19	HF	−45.23	HF	−35.76	TF
	−39.69	TF	−41.04	HF	−44.87	HF	−35.57	TF
	−39.08	TF	−41.02	HF	−44.56	TF	−35.36	TF
	−39.07	TF	−40.88	HF	−44.16	TF	−35.34	TF
**Mouse** **Alox15b mut**	−42.56	TF	−43.64	TF	−49.03	TF	−49.03	TF
	−42.56	HF	−43.28	HF	−47.95	TF	−47.95	TF
	−42.52	TF	−42.63	TF	−46.30	TF	−46.30	TF
	−42.12	TF	−42.51	TF	−46.08	TF	−46.08	TF
	−41.65	TF	−42.29	TF	−46.00	TF	−46.00	TF

**Table 3 ijms-24-10046-t003:** Average distances (Å) from the oxygen atom in the Fe(III)-OH^−^ cofactor to the closest hydrogen atom attached to C7 (H7), C10 (H10), and C13 (H13) along the molecular dynamics simulations for AA (HF or TF) at the active site of the different enzymes.

Substrate Orientation	Human						Mouse					
	WT			Mutant			WT			Mutant		
	H7	H10	H13	H7	H10	H13	H7	H10	H13	H7	H10	H13
**AA (HF)**	6.6	9.7	11.4	7.2	5.7	3.4	5.8	3.9	3.4	8.5	6.9	5.5
**AA (TF)**	6.4	3.4	4.2	6.5	4.6	4.8	8.4	5.8	3.0	7.4	4.8	3.2

**Table 4 ijms-24-10046-t004:** Average distances (Å) between the oxygen atom in the Fe(III)-OH^−^ cofactor and the closest hydrogen atom attached to C7 (H7), C10 (H10), and C13 (H13) along the molecular dynamics simulations of TF 8*S*-HETE alignment in wildtype human ALOX15B, wildtype mouse Alox15b, and Y603D+H604V mutant of mouse Alox15b, and head-first 15*S*-HETE in wildtype mouse Alox15b and D602Y+V603H mutant of human ALOX15B. n.p., not possible since neither C13 nor C10 are bisallylic methylenes in the corresponding substrates, n.c., such MD simulations have not been performed.

Substrate and Orientation	Human						Mouse					
	WT			mut			WT			mut		
	H7	H10	H13	H7	H10	H13	H7	H10	H13	H7	H10	H13
**15S-HETE (HF)**	n.c.	n.c.	n.c.	6.7	4.0	n.p.	5.6	4.5	n.p.	n.c.	n.c.	n.c.
**8S-HETE (TF)**	5.8	n.p.	2.8	n.c.	n.c.	n.c.	6.9	n.p.	4.5	6.3	n.p.	5.2

**Table 5 ijms-24-10046-t005:** Impact of reaction conditions on the rate of EPA oxygenation and the product pattern of human and mouse wildtype ALOX15B orthologs. Aliquots of bacterial lysis supernatants (5 µL) of recombinant ALOX15B preparations (wildtype human and mouse ALOX15B) were incubated for 5 min at different conditions (pH 6.4, 7.4, 8.4, 100 µM EPA, room temperature; temperatures 15 °C, 25 °C, 35 °C, 100 µM EPA, room temperature; EPA concentrations 50 µM, 100 µM, 200 µM, pH 7.4, room temperature). Sample workup and RP-HPLC analysis were carried out as described in the legend to Figure 4B. Three independent incubations were carried out under each condition (*n* = 3) and means ± SD are given.

**pH Dependence**			**Product Pattern**	
**Enzyme**	**pH**	**rel. Activity (%)**	**15-HEPE**	**8-HEPE**
Human ALOX15B	6.4	77.1 ± 3.6	100.0 ± 0.0	0.0 ± 0.0
	7.4	100.0 ± 6.3	100.0 ± 0.0	0.0 ± 0.0
	8.4	98.4 ± 6.6	100.0 ± 0.0	0.0 ± 0.0
Mouse Alox15b	6.4	100 ± 2.6	2.6 ± 0.5	97.4 ± 0.5
	7.4	62.1 ±1.0	1.5 ± 0.1	98.5 ± 0.1
	8.4	65.9 ± 10.4	3.8 ± 0.1	96.2 ± 0.1
**Temperature Dependence**			**Product Pattern**	
**Enzyme**	**Temperature (°C)**	**rel. Activity (%)**	**15-HEPE**	**8-HEPE**
Human ALOX15B	15	47.6 ±2.4	100.0 ± 0.0	0.0 ± 0.0
	25	73.7 ± 6.1	100.0 ± 0.0	0.0 ± 0.0
	35	100.0 ± 8.5	100.0 ± 0.0	0.0 ± 0.0
Mouse Alox15b	15	9.8 ± 0.9	5.8 ± 0.5	94.2 ± 0.5
	25	54.2 ± 7.0	6.7 ± 0.6	93.3 ± 0.6
	35	100.0 ±13.4	7.2 ± 1.3	92.8 ± 1.3
**Substrate Concentration Dependence**			**Product Pattern**	
**Enzyme**	**EPA Concentration (µM)**	**rel. Activity (%)**	**15-HEPE**	**8-HEPE**
Human ALOX15B	50	27.0 ± 3.6	100.0 ± 0.0	0.0 ± 0.0
	100	70.6 ± 0.8	100.0 ± 0.0	0.0 ± 0.0
	200	100.0 ± 10.9	100.0 ± 0.0	0.0 ± 0.0
Mouse Alox15b	50	11.6 ± 1.7	32.5 ± 1.3	67.5 ± 1.3
	100	88.7 ± 2.0	15.1 ± 1.2	84.9 ± 1.2
	200	100.0 ± 14.6	5.1 ± 0.5	94.9 ± 0.5

## Data Availability

All data generated or analyzed during this study are included in this published article. Original experimental raw data can be obtained upon request from K.R.K. (in vitro experiments) and A.G.L. (in silico experiments).

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
