# Peer review of "Functional Characterization of Mouse and Human Arachidonic Acid Lipoxygenase 15B (ALOX15B) Orthologs and of Their Mutants Exhibiting Humanized and Murinized Reaction Specificities"

_ijms, 2023, doi:10.3390/ijms241210046_

Round 1
Reviewer 1 Report
Specific comments for the authors
The similarity of the article with some studies in the literature (30%, iThenticate software) was found, and some parts need to be rearranged.
The importance of the study should be stated more clearly in the paragraph before the last part of the Introduction. Thus, the reader can see the purpose and importance of this study more clearly.
Line 214, “previous finding” should be referenced.
Line 253 should be (Figure 3A, B) instead of “(Figure 3A+B)”.
Line 329, A and B given in Fig 6 are not included in the explanation part. (A) should be written instead of “upper” and (B) should be written instead of “lower”.
Line 339, “Table 100. ns each).” There is an error in this part. Please review again.
Line 474, “(Figure 11A)” should be written at the end of the sentence
Line 476-479, “(Figure 11B)” should be written at the end of the sentence
Lines 558-623, In the discussion section, the results of the study should be supported by different references. References are not given in the discussion.
Author Response
Dear editor,
on behalf of all co-authors, I should like to thank the reviewers for critically reading of our ms and for their comments, which have been considered for preparation of the revised version of the paper. We addressed all their comment on the point-to-point basis in this rebuttal letter and labeled the modified paragraphs of the text in the labeled version of the ms, which we are submitting as supplemental file.
Reviewer 1:
Comment 1 of reviewer 1: The similarity of the article with some studies in the literature (30%, iThenticate software) was found, and some parts need to be rearranged.
Response of authors: We also checked the original version of the ms with the iThenticate software and found that the Materials and methods section involved text with a relatively high degree of similarity to other publications of our research group. This is not really surprising since most these methods have been used before in other studies. In the other parts of the ms the degree of textual similarity was much lower. To address this point of the evaluation report we rephrased critical parts of the ms.
Comment 2 of reviewer 1: The importance of the study should be stated more clearly in the paragraph before the last part of the Introduction. Thus, the reader can see the purpose and importance of this study more clearly.
Response of authors: We followed the advice of the reviewer and included a paragraph into the Introduction that specifically stresses the aim of this study (line 111-120).
Comment 3 of reviewer 1: Line 214, “previous finding” should be referenced.
Response of authors: The corresponding reference (25) is given (line 224).
Comment 4 of reviewer 1: Line 253 should be (Figure 3A, B) instead of “(Figure 3A+B)”.
Response of authors: We corrected the text according to the suggestion of the reviewer (line 283).
Comment 5 of reviewer 1: Line 329, A and B given in Fig 6 are not included in the explanation part. (A) should be written instead of “upper” and (B) should be written instead of “lower”.
Response of authors: We corrected the figure legend according to suggestion of the reviewer and synchronized the text (line 341 and 343).
Comment 6 of reviewer 1: Line 339, “Table 100. ns each).” There is an error in this part. Please review again.
Response of authors: The text was slightly corrupted during the formatting process and this mistake was corrected during revision (line 352 – 356).
Comment 7 of reviewer 1: Line 474, “(Figure 11A)” should be written at the end of the sentence.
Response of authors: We rephrases the corresponding paragraph of the text to explain the experimental data shown in Figure 11 in more detail (line 497-505).
Comment 8 of reviewer 1: Line 476-479, “(Figure 11B)” should be written at the end of the sentence
Response of authors: As indicated above, the corresponding paragraph of the text explaining the experimental data shown in Figure 11 was reworded.
Comment 9 of reviewer 1: Lines 558-623, In the discussion section, the results of the study should be supported by different references. References are not given in the discussion.
Response of authors: This statement of the reviewer is not entirely correct. All three paragraphs of the Discussion include references. However, in response of this criticism we include more refences into the Discussion section.
Reviewer 2 Report
The kinetics and substrate selectivity of the wild type and mutant ALOX15B orthologs are known to be sensitive to many experimental variables. In addition to fatty acid structural features that are the focus here, these include temperature, incubation time, enzyme concentration, the physical state of substrate(s), substrate-dependent activation and inactivation, and the levels of oxidants and reductants.
The need to control for such factors is illustrated by results in Fig 11, which shows large impacts of incubation time and enzyme level for reaction of mouse WT with arachidonate. However, similar data is not presented for the other three recombinant proteins and five substrates used here. It's quite possible that a different choice of supernatant concentration or incubation time would present a very different comparative picture. Making interpretation of the results even more difficult are the apparent lack of controls for many of the other variables listed above, and the incomplete description of many components in the reaction mixtures.
Overall, the targeted structure-function issues are interesting but serious gaps in the experimental design and procedural details need to be addressed to test alternate interpretations and draw reliable conclusions.
Author Response
Dear editor,
on behalf of all co-authors, I should like to thank the reviewers for critically reading of our ms and for their comments, which have been considered for preparation of the revised version of the paper. We addressed all their comment on the point-to-point basis in this rebuttal letter and labeled the modified paragraphs of the text in the labeled version of the ms, which we are submitting as supplemental file.
Response of authors: The reviewer is correct when stressing that the reaction specificity and the product pattern of lipoxygenase isoforms may depend on the experimental conditions, such as enzyme concentration, temperature, pH etc. In fact, experiments with soybean LOX1 indicated that pH alterations impact the reaction specificity of this enzyme with linoleic acid as substrate (ref. 32). On the other hand, previous results from our group indicated that mouse and human ALOX15B orthologs exhibit similar reaction specificities when the pH of the incubation mixture was modified in the near physiological range (ref. 33). To address this point of criticism of reviewer 2 were performed extensive additional experimentation comparing the composition of the product patterns of EPA oxygenation by wildtype mouse and human ALOX15B at different pH-values, different temperatures and different substrate concentrations. The results of these experiments are summarized in the new Table 5, are explained in the Results section 2.11 and are discussed in the Discussion section 3.3.
Reviewer 3 Report
The manuscript by Kakularam et al. characterizes the mechanistic basis for the catalytic specificity of the mammalian ALOX15 enzyme orthologs. In particular, the authors focused on a pair of residues previously identified in human/mouse ALOX15 enzymes (human Asp602-Val603 and mouse Tyr603-His604) critical for directing their positional catalytic specificities of PUFA oxygenation. Using an identical mutagenesis approach to the previous publication, the authors confirmed that swapping these residues between human and mouse enzymes produces a swapping of the enzyme catalytic preference for the position of the carbon double bond in the long-chain PUFAs. Using molecular docking, the authors further predict how these residues facilitate the directional binding of PUFA substrates to the enzyme catalytic pocket to achieve catalytic specificity by human and mouse enzymes. Despite extensive effort, the information generated in the current study is mainly conformational, and some methodologies may not be appropriate. The manuscript is not recommended for publication in its current form.
Major concerns:
1. Authors describe the loss of catalytic activity of the mutant enzyme in purification (Line 146-147). it is unclear if the purified mutant enzyme aggregates out of the solution or if a soluble mutant enzyme loses activity. This needs to be clarified. These phenomena suggest bacterial overexpression system may be incompatible with the folding/maturation of the mutant enzyme. If so, a eukaryotic overexpression system should be considered.
2. endpoint analysis provides valuable information in Fig 2 but is inconclusive. Suggest characterizing enzyme kinetics to determine substrate preference.
3. The information on statistical analysis is incomplete in Figure 2, Figure 9, and Figure 11A.
Minor concern:
It would be better applicable to the general audience if the information on the biological importance of the enzymes and their differential products is included in the Introduction.
Author Response
Dear editor,
on behalf of all co-authors, I should like to thank the reviewers for critically reading of our ms and for their comments, which have been considered for preparation of the revised version of the paper. We addressed all their comment on the point-to-point basis in this rebuttal letter and labeled the modified paragraphs of the text in the labeled version of the ms, which we are submitting as supplemental file.
Response of authors: Judging the degree of novelty of a scientific report is always a very personal matter and there is clearly no wrong or right. However, with all due respect, the co-authors disagree with the opinion of the reviewer that the information presented in the current study is mainly confirmational. To the best of our knowledge there is currently not a single report in the literature, in which the enzyme-substrate interaction of ALOX15B variants has been studied by means of MD simulations. Thus, although these data are consistent with previous and current experimental data the results are novel and not simply confirmational. Moreover, most of the experimental data presented in this ms (12 images and 5 data tables) are novel and have not been presented before. Finally, in the Discussion (section 3.2.) we asked five specific open questions (i to v), which were answered on the basis of the experimental data provided in this study.
It is always possible to criticize the degree of novelty of a scientific report since accumulation of knowledge is a step-by step process, which is always based on previously published data. According to our opinion independent confirmation of previous conclusion are an important part of scientific progress. In 1908 Einstein introduced the concept of space curvature as part of his general theory of relativity. In 1919 Eddington first confirmed this concept by his famous eclipse experiment and his publication had an enormous impact. Meanwhile, a large number of additional “confirmational experiments” provided independent evidence for the validity of the general theory of relativity. Most of these studies have been published although they are “confirmational”.
To avoid misunderstandings, reviewers have all the right to criticize the degree of novelty of a scientific report. However, such criticisms are just a personal opinion and editors usually decide of whether a “confirmational” report worth publishing. If a ms is published more objective measures (number of reads, citation frequency) will tell of whether or not the experimental data presented in a scientific paper are appreciated by the scientific community or not.
Comment 2 of reviewer 3: Authors describe the loss of catalytic activity of the mutant enzyme in purification (Line 146-147). it is unclear if the purified mutant enzyme aggregates out of the solution or if a soluble mutant enzyme loses activity. This needs to be clarified. These phenomena suggest bacterial overexpression system may be incompatible with the folding/maturation of the mutant enzyme. If so, a eukaryotic overexpression system should be considered.
Response of authors: As indicated in the ms the D602Y+V603H mutant of human ALOX15B lost catalytic activity during our attempts to purify the enzyme but we did not observe major protein precipitation. Moreover, in Western blotting we did not detect a loss of mutant ALOX15B protein. These data suggested that the soluble enzyme was inactivated owing to unknown mechanisms and that protein precipitation may not be responsible for the loss of catalytic activity. We also attempted to express this ALOX15B ortholog in the baculovirus-insect cell system but here again, the enzyme lost catalytic activity during the purification procedure. Thus, enzyme inactivation during purification may not be related to the recombinant overexpression system.
Comment 3 of reviewer 3: Endpoint analysis provides valuable information in Fig 2 but is inconclusive. Suggest characterizing enzyme kinetics to determine substrate preference.
Response of authors: The reviewer is correct by stressing that end point analysis does not provide detailed kinetic information. The data shown in Fig. 2 simply indicate that after a three-minute incubation period more oxygenation products were formed from DHA and EPA than from AA and LA. To explore the reaction kinetics in more detail spectrophotometric measurements should be carried out but this was not possible with the crude enzyme preparations. To address the reviewers comment we modified the text accordingly (line 206-211).
Comment 4 of reviewer 3: The information on statistical analysis is incomplete in Figure 2, Figure 9, and Figure 11A.
Response of authors: Because of the limited n-numbers (Figure 2, 9) we did not perform detailed statistical analyses of the experimental raw data. We calculated means ± DS values but did not calculate the statistical significances between the different time points and the different enzyme concentrations. Detailed statistical evaluation would not provide more valuable information.
Comment 5 of reviewer 3: It would be better applicable to the general audience if the information on the biological importance of the enzymes and their differential products is included in the Introduction.
Response of authors: We follow the advice of the reviewer and inserted a brief paragraph on the biological relevance of ALOX15B orthologs into the introduction (line 72-81).
Round 2
Reviewer 2 Report
The initial review noted that the lipoxygenase activity assays being used to measure substrate and product specificities of the recombinant enzymes did not control for potential differences between enzyme-substrate pairs in their effects on "activity" for several variables: temperature, incubation time, enzyme concentration, the physical state of substrate(s), substrate-dependent activation and inactivation, and the levels of oxidants and reductants (including bacterial lysate components). The magnitude of these differences can be substantial, and such variants are not sufficiently rare to be safely ignored. The experimental design does not use the conventional comparison of kcat / Km values calculated from initial velocity measurements for individual enzyme / substrate pairs.
Of the variables above, only temperature and pH were addressed fully in the revision. Without characterization of the remaining variables, changes in "activity" due to different interactions between substrate and enzyme will be very difficult to distinguish from "activity" changes due to altered catalytic activation by peroxide, to altered self-inactivation, to altered allosteric interactions, to differences in enzyme concentration, or to differences in relevant bacterial lysate components (e.g., oxidants, reductants, protein, lipids, detergent). As is, the study raises more questions than it answers.
Author Response
Dear editor,
on behalf of all co-authors, I should like to thank the reviewers for critically reading of our ms. Reviewers 1 and 3 were obviously satisfied by the changes introduced during the first round of revision and on behalf of all co-authors I should like to extent our gratitude for their constructive collaboration.
Reviewer 2 was not satisfied with our alterations and criticized that much more experimental work is needed to test which other variables (enzyme concentration, the physical state of substrate, substrate-dependent activation and inactivation, levels of oxidants and reductants, bacterial lysate components) might impact the reaction specificity of mouse and human ALOX15B. We think that these requests are overdemanding since corresponding experiments to each of these points would exceed the frame of this study. The ms already involves 12 figures and 5 data tables and testing all the variables suggested by the reviewer and possibly also other experimental conditions on their potential effects would increase the data multiplicity even more. During the first round of revision, we selected 3 different variables (pH, temperature, substrate concentration), which according to our opinion are of major physiological relevance, and tested their impact on the reaction specificity of wildtype mouse and human ALOX15B. We feel that these experiments, the results of which are summarized in Table 5, did adequately address the critical comment of this reviewer. It was not the aim of our study to explore in detail how alterations in the reaction conditions may impact the reaction specificity of human and mouse ALOX15B orthologs and thus, the request of the reviewer to test a large number of different experimental condition exceeds the frame of this study.
In addition to these more general comments, we would like to specifically refer to the other critical remarks of the reviewer.
Comment of reviewer: The initial review noted that the lipoxygenase activity assays being used to measure substrate and product specificities of the recombinant enzymes did not control for potential differences between enzyme-substrate pairs in their effects on “activity” for several variables: temperature, incubation time, enzyme concentration, the physical state of substrate(s), substrate-dependent activation and inactivation, and the levels of oxidants and reductants (including bacterial lysate components). The magnitude of these differences can be substantial, and such variants are not sufficiently rare to be safely ignored.
Response of authors: According to our opinion this text does not correctly describe the importance of the reaction specificity of lipoxygenases in general. It gives the impression that this enzyme property is very variable and can easily be changed by alterations of the reaction conditions. This is, however, not the case. For most ALOX isoforms the reaction specificity with a given substrate is well defined and alterations in the reaction conditions do hardly alter this enzyme property. There may be minor modifications of the reaction specificity when the reaction conditions are modified in the near physiological range, but in principle the reaction specificities of ALOX-isoforms are very robust. Different reaction conditions modify the catalytic efficiencies (kcat, Km) of the enzymes but except from rare cases the reaction specificities remain unaltered. This conclusion is supported by the experimental date shown in Table 5. Here we found that the reaction specificity of human ALOX15B does not depend on pH, temperature and substrate concentration. The reviewer might argue that the pattern of the arachidonic acid oxygenation products strongly depends on the amount of enzyme added to the incubation mixture (Figure 11B). Although correct this finding does not prove that the reaction specificity of the enzyme has changed. In fact, these differences in the product pattern are a consequence of enzyme kinetics. At higher enzyme concentrations the fatty acid substrate becomes rate limiting and the primary reaction product (8-HETE) accumulates. Under these conditions the primary reaction product (8-HETE) is used as substrate for secondary oxygenation (formation of 8,15-HETE). Thus, also at high enzyme concentration the reaction specificities of the enzymes remain more or less unchanged. We introduced a short paragraph in the Discussion section (lines 700-714) to stress this point.
Comment of reviewer: The experimental design does not use the conventional comparison of kcat / Km values calculated from initial velocity measurements for individual enzyme / substrate pairs.
Response of authors: The reviewer is correct that for detailed kinetic experiments the catalytic efficiency of an enzyme is usually characterized by the kcat/Km ratio. However, such ratios can only be calculated for pure enzyme preparations since kcat values cannot be determined for crude enzymes. Our studies were carried out with crude enzyme preparations and the reasons for this are explained in the paper.
Comment of reviewer: Of the variables above, only temperature and pH were addressed fully in the revision.
Response of authors: This statement is not correct since we explored the impact of pH, T and substrate concentration (Table 5) on the reaction specificity of human and mouse ALOX15B.
Comment of reviewer: Without characterization of the remaining variables, changes in "activity" due to different interactions between substrate and enzyme will be very difficult to distinguish from "activity" changes due to altered catalytic activation by peroxide, to altered self-inactivation, to altered allosteric interactions, to differences in enzyme concentration, or to differences in relevant bacterial lysate components (e.g., oxidants, reductants, protein, lipids, detergent).
Response of authors: To the best of our knowledge there is no indication in the ALOX literature that the presence/absence of peroxide activators may alter the reaction specificity of any ALOX isoform. In fact, when we carried out our incubations of mouse and human ALOX15B in the presence of micromolar concentrations of H2O2 the reaction specificity was not altered. The reaction rate is modified but the reaction specificity is not.
Similarly, the degree of self-inactivation of rabbit ALOX15 did not alter the reaction specificity of the enzyme. In fact, the reaction specificity of partly self-inactivated rabbit ALOX15 exhibits the same reaction specificity as the fully activated enzyme.
The reviewer claimed in his evaluation report that “altered allosteric interactions” might modify the reaction specificity of ALOX isoforms but, unfortunately, he did not give a corresponding reference. According to our experience allosteric ALOX15 inhibitors did only minimally modify the reaction specificity of rabbit ALOX15. In fact, we are not aware of any literature report indicating pronounced alterations in the reaction specificity by allosteric ALOX inhibitors.
Addition of detergents alters the reaction rate of ALOX-isoforms owing to a better substrate availability. However, detergents at low concentrations do not alter their reaction specificity. We found that sodium cholate does neither modify the reaction specificity of rabbit ALOX15 nor that of mouse and human ALOX15B with arachidonic acid as substrate.
The reviewer also claimed that the reaction specificity of ALOX-isoforms may be altered by components of the bacterial lysate but here again no reference is given. There are many reports in the literature which indicate that crude recombinant lipoxygenase preparations exhibit the same reaction specificity as the purified enzymes. Moreover, a large number of cellular studies also indicate that the reaction specificity of ALOX isoforms inside cells is very similar to that of the purified enzymes. In some cases, there might be minor differences but these differences might be related to structural changes the ALOX-isoforms experience during the purification procedure. A good example for this is purification of native rabbit ALOX15 (FEBS Lett. 1983 Mar 21;153(2):353-6. doi: 10.1016/0014-5793(83)80641-3). Here the reaction specificity of the enzyme was very similar all stages of the multistep purification procedure.
Comment of reviewer: As is, the study raises more questions than it answers.
Response of authors: In this paper we asked the question whether humanization/murinization of the reaction specificity of mouse and human ALOX15B is restricted to arachidonic acid as substrate or whether this may also be the case with other polyenoic fatty. Moreover, in the modeling part of the ms we explored the molecular basis of the differential reaction specificity of mouse and human ALOX15B. In other words, we asked two specific questions and answered both of them in detail. The reviewer is correct that the experimental data obtained in our study opened a number of additional questions. But this is not a limitation but rather a strength of our paper. In fact, in most research reports one or two specific questions are answered but each answer opens several new questions. This is the way science moves forward.
We are submitting a labeled version of the rerevised ms, in which the textual alterations are indicated by yellow background. We hope that we have answered all concerns of the reviewer and that the revised version of the ms is now acceptable for publication.
On behalf of all co-authors
H. Kuhn
Round 3
Reviewer 2 Report
The distinction drawn in the author's detailed comments between the effects of experimental conditions on reaction rates versus those on reaction specificity is very useful, in that it points to a potential source of confusion for readers that could be fixed with minor edits.
Globally replacing the term "reaction specificity" with "product specificity" (and associated text edits) would help readers distinguish between comparisons of substrate specificity (eg, Figs 2 and 9) where interpretations remain contentious without kcat/Km data, and product or positional specificity comparisons, whose validity does not hinge on reaction rate measurement, and which are the main topic of this study.
Author Response
We follow the advice of the reviewer and replaced the phrase "reaction specificity" with "product specificity" ("product pattern", "product mixture") throughout the text. The term "product specificity" does not appear any more in the final version of the ms.